# Floristic diversity and its relationships with human land use varied regionally during the Holocene

Jonathan D. Gordon [1,2,3] ✉, Brennen Fagan [1,4], Nicky Milner [1,3] & Chris D. Thomas [1,2]

Humans have caused growing levels of ecosystem and diversity changes at a global scale in recent centuries but longer-term diversity trends and how they are affected by human impacts are less well understood. Analysing data from 64,305 pollen samples from 1,763 pollen records revealed substantial community changes (turnover) and reductions in diversity (richness and evenness) in the first ~1,500 to ~4,000 years of the Holocene epoch (starting 11,700 years ago). Turnover and diversity generally increased thereafter, starting ~6,000 to ~1,000 years ago, although the timings, magnitudes and even directions of these changes varied among continents, biomes and sites. Here, modelling these diversity changes, we find that most metrics of biodiversity change are associated with human impacts (anthropogenic land-cover change estimates for the last 8,000 years), often positively but the magnitudes, timings and sometimes directions of associations differed among continents and biomes and sites also varied. Once-forested parts of the world tended to exhibit biodiversity increases while open areas tended to decline. These regionally specific relationships between humans and floristic diversity highlight that human–biodiversity relationships have generated positive diversity responses in some locations and negative responses in others, for over 8,000 years.

Human-driven biotic homogenization and the erosion of biological diversity are widely reported in the scientific literature[1,2], as articulated by the Intergovernmental Science-Policy Platform on Biodiversity and Ecosystem Services: "Biodiversity… is declining faster than at any time in human history", with projected continuing declines[3]. However, there is continuing debate. For example, although major land-use changes do reduce the number of species per unit area[1], analyses of time-series data (repeatedly sampled locations) over the last ~70 years commonly find that local diversity (richness) gains have, on average, balanced losses, despite accelerating rates of community change and considerable variation in trends at individual sites[4–7]. These different perspectives raise the question of how the diversity of ecological communities has

changed during the deeper history of human development, whether different metrics of biodiversity change reveal similar or different trends and how changing human land use relates to these changes.

Previous work has provided valuable insights into site-level[8,9], regional[10–13] and continental-scale[14,15] changes to certain aspects of plant diversity over multimillennial timescales but the above debate concerning contemporary biodiversity data highlights the value of considering multiple metrics of biodiversity together. Here we extend the analysis to additional metrics of biodiversity and additional data to provide a multimetric analysis of Holocene floristic diversity changes across all continents, except Antarctica, and to provide biome-level analyses for North America and Europe. We relate these diversity

[1]Leverhulme Centre for Anthropocene Biodiversity, University of York, York, UK. [2]Department of Biology, University of York, York, UK. [3]Department of Archaeology, University of York, York, UK. [4]Department of Mathematics, University of York, York, UK. ✉e-mail: jonny.gordon@york.ac.uk

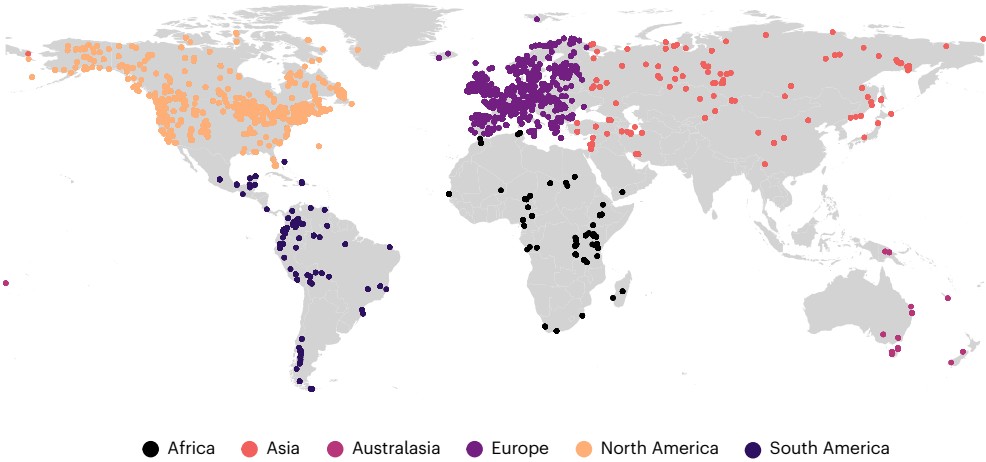

Africa • Asia • Australasia • Europe • North America • South America

**Fig. 1 | Fossil pollen records included in these analyses.** The records included in these analyses are coloured by 'continent'.

metrics to estimates of human-induced land-cover change to evaluate how human land use may have influenced diversity trends over the period of most substantial human modification to the surface of Earth. More specifically, we relate the 'Krumhardt Kaplan version 2010'[16,17] (hereafter, KK10) estimates of anthropogenically induced land-cover change (ALCC) to all of our diversity metrics from 8,000 to 100 calibrated years before radiocarbon present (hereafter cal yr BP, with 'radiocarbon present' being 1950 CE), the period over which the KK10 ALCC estimates and our pollen data overlap.

We analysed data from pollen records, which contain information relating to plant diversity and their associated chronological data ranging from 11,700 to 100 cal yr BP, only considering records with several Holocene pollen samples and date estimates (Extended Data Fig. 1 gives a methodological overview). These filtered data constitute 1,763 pollen records (Fig. 1) with an average of ~36 pollen samples per record, totalling 64,305 pollen samples (Extended Data Fig. 2). We estimated six measures of pollen-type diversity: two describing changes in community composition through time (turnover), two describing diversity in each sample (richness and evenness) and two describing how communities differed from one another across space (heterogeneity) (Figs. 2 and 3 and Extended Data Fig. 3). We relate these computed diversity values to the KK10 ALCC scenario in models at the continental scale (Fig. 4) and also at the biome scale for Europe and North America (Fig. 5).

To control for sample size differences, we created 1,000 resamples of the pollen dataset by sampling (without replacement) 300 pollen grains from each pollen sample (generating resampled datasets of up to 63,420 samples, each consisting of 300 grains). To ensure consistency across records[18], we produced an updated chronology for each of the 1,763 pollen records using a Bayesian age–depth model and assigned a random draw from the posterior distribution of the corresponding age–depth model for each each pollen record to each resampled pollen dataset, thereby incorporating the temporal uncertainty inherent in age–depth modelling and resampling procedures into our analyses (Methods). For each resample, we calculated the diversity metrics for each continent and also for the World Wildlife Fund terrestrial ecoregions[19] (hereafter, biomes) in North America and Europe, the two continents with sufficient data to undertake biome-level analyses. For spatial heterogeneity, we controlled for the sample size of pollen records within each bin by bootstrap resampling to the lowest number of pollen records present across all bins and regions considered.

These analyses produced 1,000 time series of each metric for each continent and for each biome in Europe and North America. We then fitted generalized additive models (GAMs) to the diversity metrics of each time series, with a separate smooth for each continent/biome.

We represent the cohort of 1,000 model predictions for each continent (Fig. 2 and Extended Data Fig. 3a) and biome (Fig. 3 and Extended Data Fig. 3b) as medians and interquartile ranges.

## Results

### Variation in diversity change among continents

Turnover quantifies the rate at which the composition of the vegetation (from pollen samples) at each site was changing through time (Methods). For each pollen record, we quantified the compositional turnover between successive samples along the record (that is, we measured compositional change within sites), adjusted for the time interval between sequential samples using both the abundance-based Bray–Curtis (Fig. 2) and the incidence-based Jaccard measures (Extended Data Fig. 3a). These analyses reveal that the turnover of pollen types was relatively high for all continents (except Australasia and Africa) immediately following the Pleistocene/Holocene climatic transition but sampled pollen assemblages then gradually stabilized (reduced turnover) over the following 2,000 to 7,000 years, depending on the continent considered (Fig. 2). Turnover subsequently increased again in each continent (apart from in Asia and Africa) but at different times. Rates have exceeded the Pleistocene/Holocene turnover peak since ~3,000 cal yr BP in Europe and since the Mid Holocene in Australasia.

Pollen richness quantifies the number of unique pollen types per sediment subsample for a fixed number of pollen grains[20] (300 pollen grains in these analyses). Across the Northern Hemisphere, pollen richness decreased following the Pleistocene/Holocene climatic transition and the subsequent community adjustment (succession) (Fig. 2). However, the duration of the richness decline varied among continents. Following this reduction, richness increased from ~10,500 cal yr BP in Europe (Fig. 2b), ~7,500 cal yr BP in Asia (Fig. 2j) and ~6,500 cal yr BP in North America (Fig. 2f).

The picture is much less clear for the Southern Hemisphere, with South American richness decreasing to ~2,000 cal yr BP, followed by relative stasis (Fig. 2n), while Africa shows an overall positive richness trend over the first half the Holocene, followed by sharp decreases from ~5,000 cal yr BP to 1850 CE (Fig. 2r). Australasian richness sharply reduces following the onset of the Holocene up to ~8,000 cal yr BP, after which it increases but falls again from ~3,000 cal yr BP to 1850 CE (Fig. 2v). The differences between the richness trends for African, Australasian and South America sites (compared to the higher latitude Northern Hemisphere sites) are not unexpected: the climatic transition from Late Pleistocene glacial to Early Holocene climates manifested very differently between these regions, especially with regard to regional hydroclimatic conditions[21].

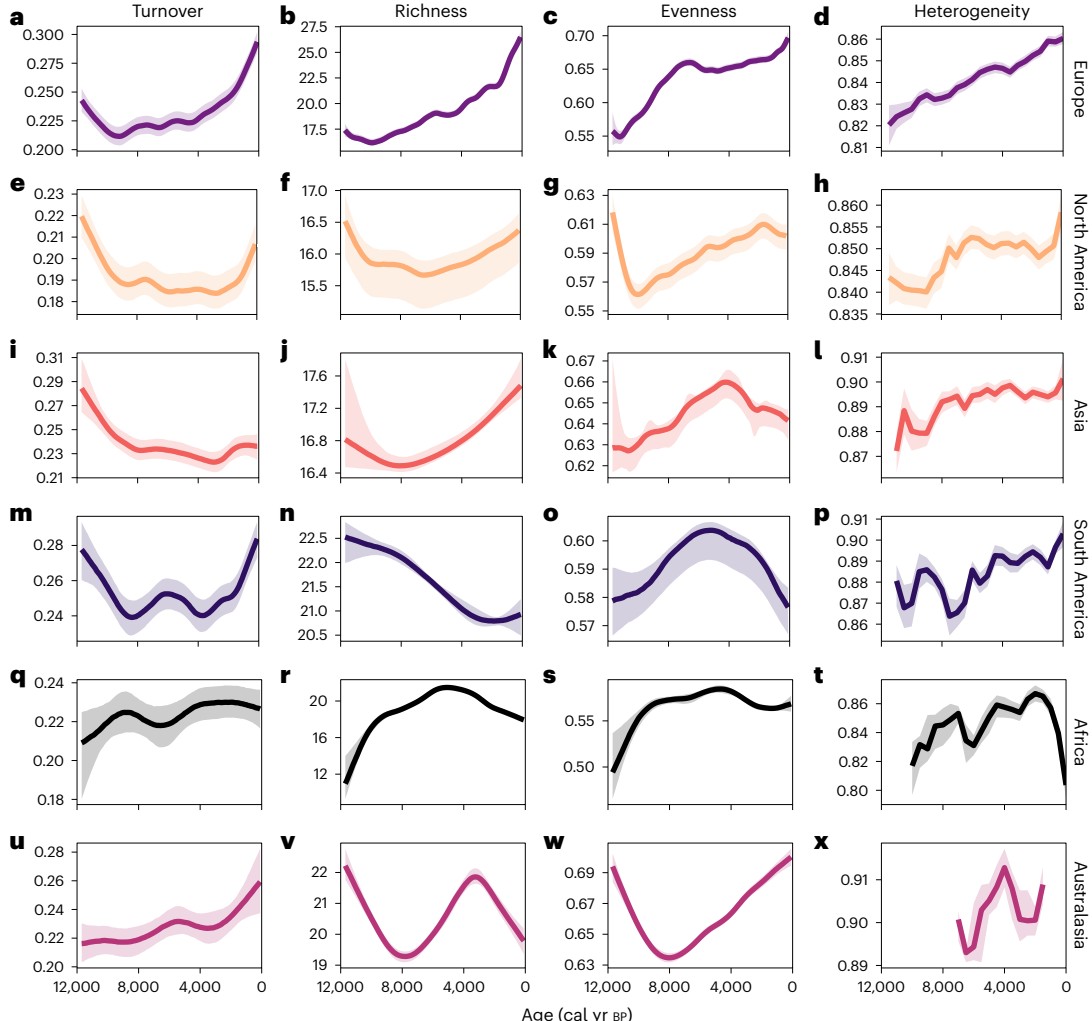

**Fig. 2 | Continental Holocene pollen diversity trends.** GAMs fit to continental Holocene pollen turnover (Bray–Curtis), richness and evenness data. Heterogeneity panels (final column) are not fitted GAMs but distributions of the resampled data (given the reduced heterogeneity sample size arising from the 500 year binning; Methods). The 1,000 resamples (modelled first by GAMs, except in the final column) are then summarized by the median (thick lines) and interquartile ranges (shaded intervals). Rows are ordered by the number of pollen samples present in each continent. Europe *n* = 39,804 (**a**–**d**), North America *n* = 16,897 (**e**–**h**), Asia *n* = 2,882 (**i**–**l**), South America *n* = 2,524 (**m**–**p**), Africa *n* = 1,648 (**q**–**t**) and Australasia *n* = 550 (**u**–**x**).

Pollen evenness measures the extent to which different pollen types are relatively equally shared among taxa (even) or dominated by one or a few pollen types (uneven). Evenness patterns vary over the first ~2,000 years of the Holocene across the Northern Hemisphere, increasing thereafter, although the patterns of change differed between the continents (Fig. 2c,g,k). South American and African evenness patterns are similar, peaking during the Mid Holocene, whereas Australasian trends increase from ~8,000 cal yr BP to the present.

Given the substantially reduced number of datapoints for Bray–Curtis spatial heterogeneity (one value per 500 year bin), we did not model the heterogeneity metric through time. Rather, we present the distribution of results through time across the 1,000 resamples in Figs. 2 and 3. The overall heterogeneity of pollen samples across space (between-site compositional differences or beta diversity) increased from the onset of the Holocene to 1850 CE in the Northern Hemisphere (Fig. 2d,h,l) and since ~7,000 cal yr BP in South America (Fig. 2p). In Africa, heterogeneity increases overall from ~9,000 to ~2,000 cal yr BP, after which there is a rapid reduction (representing an homogenization of vegetation communities across space) to 1850 CE (Fig. 2t). Jaccard-based heterogeneity trends are similar to those of Bray–Curtis across all continents except Europe, which shows a reduction in

heterogeneity until ~7,000 cal yr BP followed by an increase to 1850 CE (Extended Data Fig. 3a). It is worth noting that heterogeneity (dissimilarity) values are consistently high because of the environmental and geographic scale of continents (the floras remain distinct in different locations, despite increasing turnover).

Across all four modelled metrics (and >95% of the resamples), including a separate smooth for each continent resulted in an Akaike information criterion (AIC) reduction by >2 compared with a model fit to the data with no continental subdivision (that is, improved the model fit[22]; Supplementary Table 2). This is evidence that diversity pattern changes varied among continents for these metrics.

## Variation in diversity change among biomes and sites

There is sufficiently high (temporal and spatial) density of pollen records across North America and Europe to model pollen diversity changes in finer detail: at the biome scale. There is substantial variation in Holocene biome diversity trends within and between continents (Fig. 3), especially within North America (Fig. 3, solid lines), whereas European biome trends (Fig. 3, dashed lines) are much more consistent. Notably, Holocene compositional turnover increases across all biomes in both Europe and North America (except temperate grasslands) in

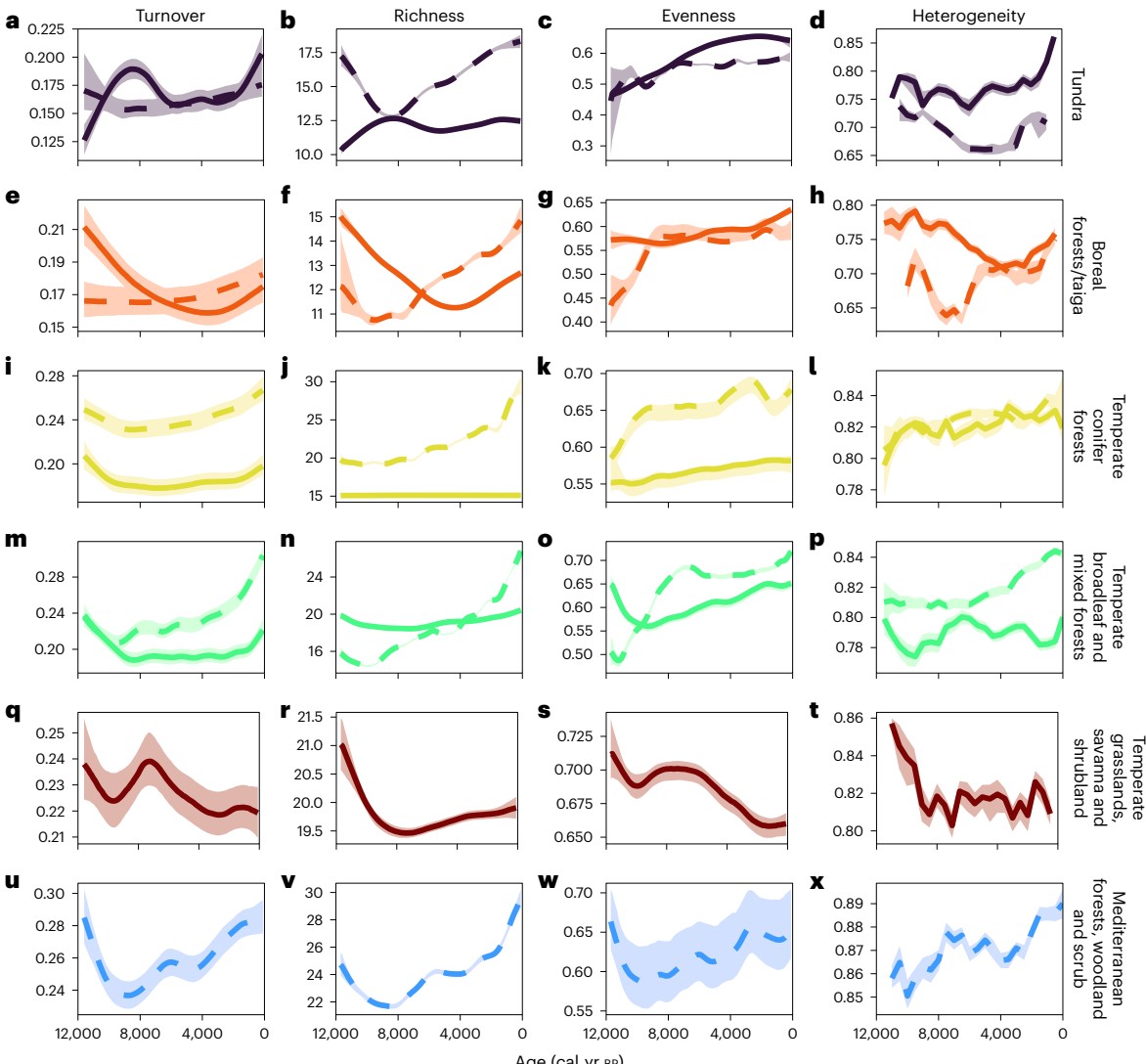

**Fig. 3 | European and North American Holocene pollen diversity trends by biome.** GAMs fit to North American (solid lines) and European (dashed lines) Holocene pollen turnover (Bray–Curtis), richness and evenness data. Heterogeneity panels (Bray–Curtis; final column) are not fitted GAMs but distributions of data (given the reduced heterogeneity sample size arising from the 500 year binning; Methods). Holocene pollen diversity data are shown for each biome that is represented by at least 20 pollen records. The 1,000 resamples (modelled first by GAMs, except in the final column) are then summarized by median (thick lines) and interquartile ranges (shaded intervals). Rows are ordered approximately by latitude. North America (NA) tundra $n = 1,130$, Europe (EU) tundra $n = 1,416$ (**a**–**d**), NA boreal forests/taiga $n = 2,048$, EU boreal forests/taiga $n = 2,018$ (**e**–**h**), NA temperate conifer forests $n = 4,242$, EU temperate conifer forests $n = 5,469$ (**i**–**l**), NA temperate broadleaf and mixed forests $n = 6,645$, EU temperate broadleaf and mixed forests $n = 26,235$ (**m**–**p**), NA temperate grasslands, savannas and shrublands $n = 1,691$ (**q**–**t**), EU Mediterranean forests, woodland and scrub $n = 3,750$ (**u**–**x**).

the later Holocene, although the uptick is more pronounced and earlier, in Europe. The Jaccard-based turnover results (Extended Data Fig. 3b) show qualitatively similar patterns to the Bray–Curtis analysis for most European biomes (Fig. 3) but differ (and are noisier) for North American ones: Jaccard turnover reduced over the Holocene in North America apart from in the temperate grasslands and boreal region (Extended Data Fig. 3b). Bray–Curtis turnover (which includes pollen-type abundance changes as well as in pollen identities) is positively correlated with Jaccard turnover (which only includes pollen-type presence–absence) for three out of five European biomes; but the two metrics are not correlated for North American biomes (Extended Data Fig. 4a). This lack of correlation implies greater contributions of relative abundance differences (of taxa still present) to community composition changes in North America compared to Europe.

Focussing on richness, two out of five North American biomes show declining richness over the Holocene (boreal forests/taiga;

temperate grasslands, savanna and shrubland; Fig. 3f,r), whereas the remainder either remain constant (temperate conifer forests; Fig. 3j) or increase (tundra; temperate broadleaf and mixed forests; Fig. 3b,n). In contrast, European richness increases across all biomes over the same period. These divergent continental richness responses between the same biomes are particularly interesting given the correlations between compositional turnover and richness: in both Europe and North America, richness and turnover are positively correlated across most biomes (Extended Data Fig. 4a).

The analyses indicate, for both continents, that inclusion of the biomes separately increases the overall model fit (judged by AIC; Supplementary Table 3) for richness, evenness and both measures of turnover. Thus, the varying trends represent statistically identifiable differences among biomes, rather than sampling error. Trends vary among regions-within-continents (Fig. 3) as well as between continents (Fig. 2).

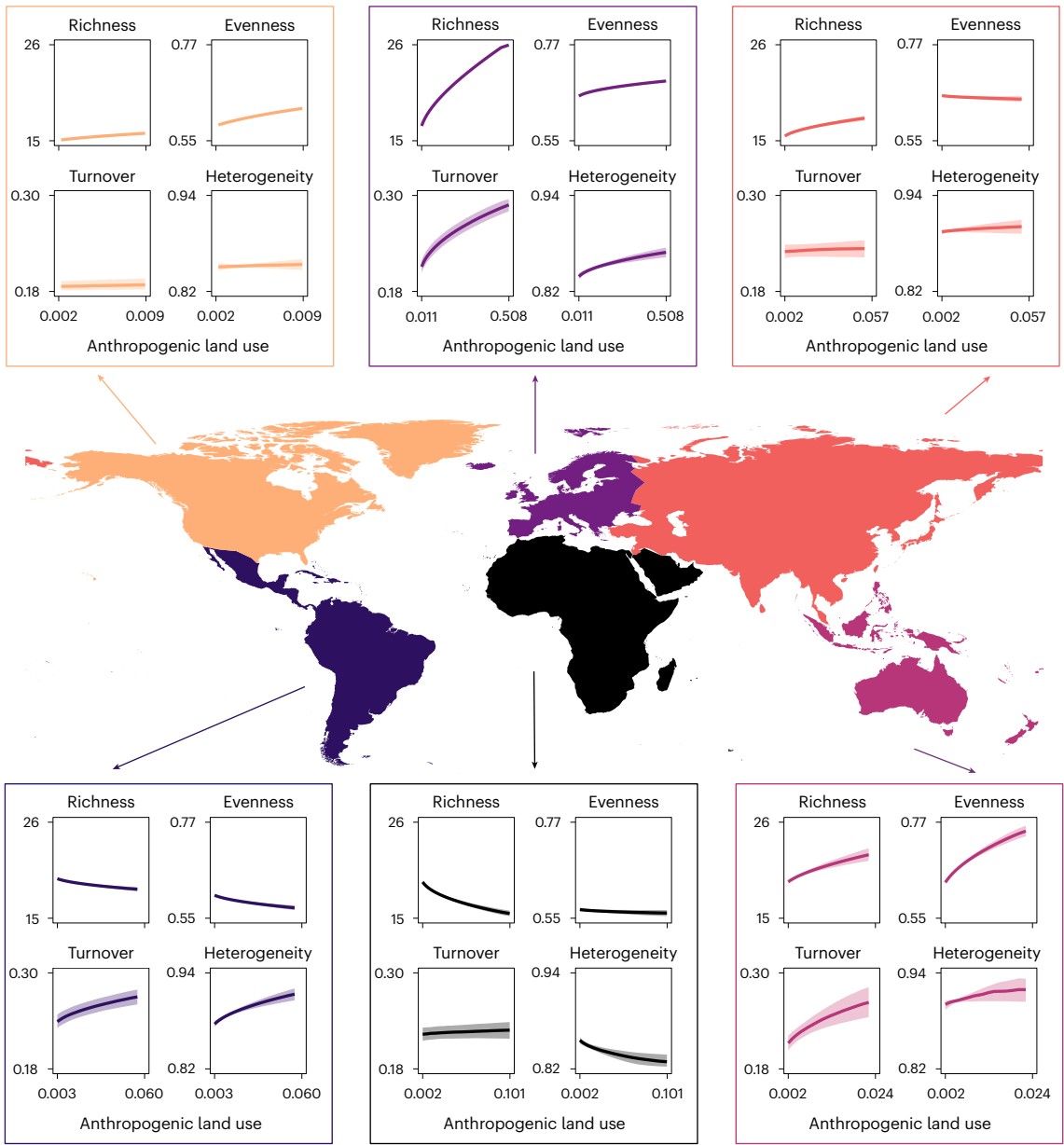

**Fig. 4 | Fitted relationships between each diversity metric and anthropogenic land use for each continent for data encompassing the period 8,000–100 cal yr BP.** Lines represent median fits across 1,000 resamples; shaded intervals represent interquartile ranges across 1,000 resamples. Note that anthropogenic land use is back transformed and axis ranges differ between continents. Heterogeneity is calculated by Bray–Curtis. Europe $n$ = 32,793, North America $n$ = 13,025, Asia $n$ = 2,407, South America $n$ = 2,169, Africa $n$ = 1,458 and Australasia $n$ = 431.

The overall continental and biome trends are smoothed fits, based on many pollen records, which vary greatly in their individual biological and human histories. As such, the individual trajectories at specific locations (single pollen records) are variable. For illustration, we show the slopes of linear model fits to individual pollen record diversities over three equal periods encompassing the Early Holocene, Mid Holocene and Common Era (Extended Data Figs. 5 and 6). For all four metrics and three time periods considered, some pollen records exhibit increases and others decreases within all continents and biomes but the proportions of increases and decreases vary. These results highlight that there is a shifting mosaic of spatially and temporally variable diversity patterns across the entire Holocene, such that the observed overall continental and biome changes in Figs. 2 and 3 arise from changing proportions of records showing increases or decreases of various magnitudes, rather than all records behaving in a similar manner. This mirrors modern biodiversity time-series data, where temporal trends (and lack of trends) are constructed of some sites that have increased in diversity and others where diversity has declined[4–7].

**Diversity changes in relation to human land-cover changes**
To understand how these observed continental- and biome-scale diversity patterns relate to human land-use changes, we modelled each of the diversity metrics against the global KK10 anthropogenic land-cover change scenario (ALCC)[16,17]. For each continent and biome, we generated an annually resolved ALCC time series from the gridded KK10 scenario by calculating the mean ALCC value across all grid cells contained in each region over the period of data availability, 8,000–100 cal yr BP. We regressed each diversity metric against the regional square-root transformed ALCC values in generalized linear mixed effects models and present the results as fits for each continent in Fig. 4 and for the biomes considered in Fig. 5.

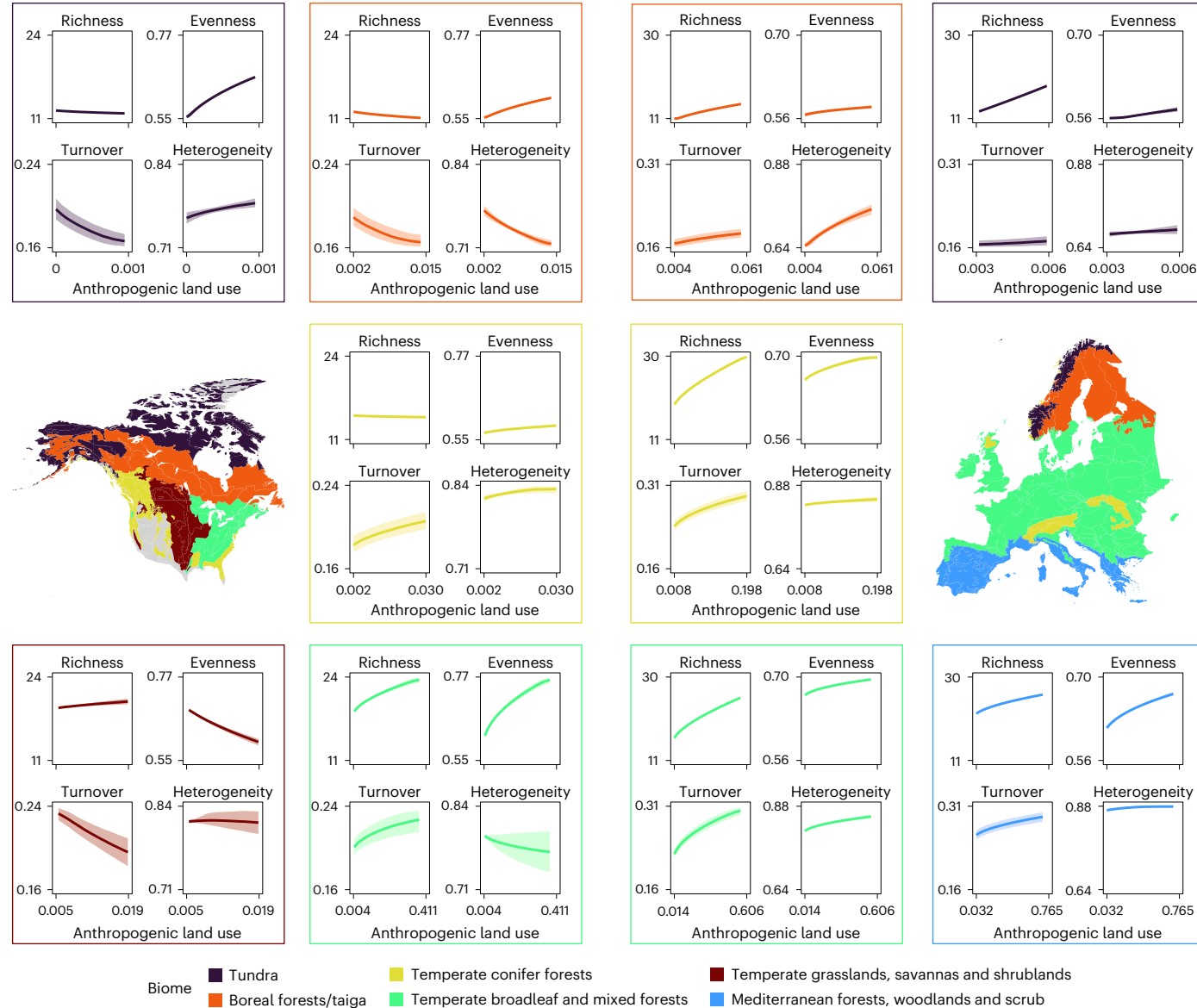

**Biome**
- Tundra
- Boreal forests/taiga
- Temperate conifer forests
- Temperate broadleaf and mixed forests
- Temperate grasslands, savannas and shrublands
- Mediterranean forests, woodlands and scrub

**Fig. 5 | Fitted relationships between each diversity metric and anthropogenic land use for North American and European biomes for data encompassing the period 8,000–100 cal yr BP.** Relationships shown are for those biomes within North America (left-hand panels) and Europe (right-hand panels) that met the inclusion criteria, encompassing the period 8,000–100 cal yr BP (Methods). Lines represent median fits across 1,000 resamples; shaded intervals represent interquartile ranges across 1,000 resamples. Note that anthropogenic land use is back transformed. Heterogeneity is calculated by Bray–Curtis. Diversity axis ranges are consistent within but not between continents. Grey land surface areas are not considered. NA tundra n = 872, EU tundra n = 1,016, NA boreal forests/taiga n = 1,725, EU boreal forests/taiga n = 1,704, NA temperate conifer forests n = 3,204, EU temperate conifer forests n = 3,816, NA temperate broadleaf and mixed forests n = 5,002, EU temperate broadleaf and mixed forests n = 22,666, NA temperate grasslands, savannas and shrublands n = 1,242, EU Mediterranean forests, woodland and scrub n = 2,834.

Community turnover is positively associated with ALCC in every continent, although the effect size is small in Asia and North America relative to the other continents. In contrast, the relationships of richness, evenness and both metrics of spatial heterogeneity with ALCC vary by continent (see Extended Data Fig. 7 for Jaccard turnover and heterogeneity results). In the Northern Hemisphere, all three measures of diversity (aside from Asian evenness) increase with increasing fractions of ALCC, with the strongest positive trends in Europe. Trends are generally weakest in North America; the reason for this weaker continental-scale trend can be explained by the biome-specific differences in Fig. 5. Southern hemisphere trends are more mixed. In Africa, all metrics other than turnover reduce as ALCC values increase. In South America, richness and evenness decrease with increasing ALCC, although heterogeneity increases. In contrast to the other Southern

Hemisphere general trends, all diversity metrics increase in association with ALCC in Australasia.

We also investigated the relationships between diversity and ALCC for the European and North America biomes that met our inclusion criteria (Fig. 5; see Extended Data Fig. 7 for Jaccard turnover and heterogeneity results).

In Europe, turnover and all diversity metrics are positively associated with ALCC, with the strongest effects in the mid-latitude temperate broadleaf and conifer forests (Fig. 5, green and yellow biomes, respectively). While associations are weaker in the extreme latitudes, they remain positive across each biome. In North America, turnover and diversity are generally positively associated with ALCC in the lower-latitude temperate broadleaf and conifer forests, although in the broadleaf forests an increasing human land-cover change is

associated with a homogenization of the floristic community. At higher latitudes (the boreal forests/taiga and the tundra) and in the Great Plains, diversity is largely negatively associated with ALCC, although richness does slightly rise with ALCC increases in the Plains. These positive associations in the North American forested biomes and the negative associations in the other three biomes are probably driving the flat or weak continent-scale North American trends in Fig. 4, given that opposing patterns from different biomes appear to be negating each other.

### Interpreting pollen diversity trends

We note several important caveats. These analyses represent the diversity of pollen samples within pollen records, rather than of an inventory of plant taxa in proportion to their abundances in the surrounding landscape. Differential pollen production/dispersal and taxonomic resolutions, for example, provide challenges to the interpretation of pollen diversity results—especially richness—in relation to plant communities[23,24], which we have endeavoured to mitigate against and demonstrate robustness to, as far as possible through our data selection, analytical approaches and sensitivity analyses (see below and Methods). Despite this uncertainty, monitoring and simulation studies provide evidence for a positive relationship between pollen-type richness and the richness of the plant communities that the pollen represents[25–28]. Pollen samples are indeed samples and currently they provide the best source of information on past vegetation diversity (including richness).

Our conclusions hold for the records from which pollen samples that met our 'data quality' criteria were available (Fig. 1) and hence the results should be interpreted as averaged across records that met our inclusion criteria. Pollen deposition environments are mainly water bodies and wetlands and these are non-random locations and environments within each continent. Increasing rates of ecosystem change and potentially diversification, may also have operated in drier (generally non-depositional) environments[29] but this requires further empirical exploration. Variation in pollen source area and taphonomic processes between depositional environments also has the capacity to affect modelled pollen diversity trends. We investigated any potential effect of depositional environment on our results by performing leave-one-out sensitivity analyses, in which each depositional environment was excluded in turn and all trends and models were re-estimated for the remaining data (Methods). The turnover, diversity and heterogeneity trends were robust to these exclusions, indicating that no single type of depositional environment (that was atypical or had extreme effect sizes) was driving the overall patterns of biodiversity change at continental or biome scales or the associations of those trends with human land-use change (Extended Data Figs. 8–10). Finally, we emphasize that we have not analysed impacts more recently than 1850 CE to avoid comparing non-equivalent surface- and within-deposit pollen profiles. Our results therefore predate the Great Acceleration of increasing environmental change[30]. Our analysis of human impacts also extends only as far as 8,000 cal yr BP, given the KK10 land-use scenario data availability, so we cannot comment on human effects before this date.

### Discussion

Overall, rates of community change (turnover) have accelerated for the last ~1,000 to ~9,000 years across most continents and biomes within North America (except temperate grasslands) and Europe. This temporally asynchronous but qualitatively consistent turnover trend is associated with a diverse range of responses in the richness, evenness and spatial heterogeneity of pollen samples over the same periods.

From 11,700 to ~8,000 cal yr BP, measures of diversity and community change are likely to reflect extended responses to the global Late Pleistocene to Holocene climatic transition. Thereafter, between ~8,000 and 4,000 cal yr BP, some level of stasis is a common feature of many (but not all) continental and biome trends; at the site level, similar numbers of sites exhibited increases in diversity as declines (Extended Data Figs. 5 and 6). This does not mean that all communities were stable but that the overall multisite ensemble of Mid Holocene floristic communities appear to have been in a relatively steady state (in terms of average diversity metrics) during this period. From 4,000 cal yr BP to 1850 CE, however, there has been an increase in the rates of community change across all continents (except Africa) and most biomes in Europe and North America. This increased turnover is associated with increased diversity in some locations but declines in others, as exemplified by positive associations between pollen richness and turnover in Europe, North America (for Bray–Curtis but not for Jaccard) but negative or uncorrelated richness–turnover relationships trends in Asia, Australasia and South America (Extended Data Fig. 4b). In addition to these continental-scale trends, individual sites exhibit considerable local variability (Extended Data Figs. 5 and 6), which probably reflects diverse human and other drivers playing out in different environmental circumstances; for example in relation to local geologies, soils, biotas and, potentially, human societal developments.

Between 8,000 and 100 cal yr BP we find that there is a positive association between turnover and ALCC at the continental scale, although the levels of turnover vary (Fig. 4). Given the increasing human environmental footprint and disturbance from the Mid Holocene onwards[31], an increase in the turnover of floristic communities from this period onwards is perhaps expected a priori. Despite this, other metrics show interesting and different responses between and within continents. In Europe, for example, extensive deforestation and harvesting of broadleaf woodland, which accelerated during the Late Bronze and Iron Ages[32–34] (after ~4,000 cal yr BP), would have generated more habitats for plants to exploit, a landscape modification that coincided with concurrently higher rates of turnover (Fig. 2a) and increasing richness (Fig. 2b) (see Extended Data Fig. 4 for intracontinental metric correlations); and both metrics are positively associated with changes in ALCC over the same period (Figs. 4 and 5). These increases in site-level diversities across Europe did not result in human-driven homogenization of the floristic communities at the continental (Fig. 4) or biome scales (Fig. 5), highlighting that the potential homogenization of plant communities indicated from recent observational studies[35] occurred primarily since 1850 CE. Indeed, across all continents considered, it is only in Africa where increasing human land use is associated with a homogenization of floristic communities (Fig. 4). Most African pollen records in this study are from the tropical savannahs, where agropastoralism was the primary model of subsistence (compared to plant cultivation in other continents) over much of the later Holocene[36,37]. It is possible that this subsistence mode, intensifying after ~3,000 cal yr BP (ref. 38), led to the observable opposite trend in heterogeneity in the African Late Holocene through regionally intensive grazing and other livestock-associated pressures.

Recent monitoring studies have revealed an acceleration in the compositional turnover of ecological communities over the last ~50 to ~100 years but a variety of responses for richness changes in the same assemblages, leading to no overall average richness change[4,5,39]. Our results over much longer timescales (Figs. 2 and 3) resonate with these conclusions (accelerating turnover, underpinned by heterogeneous mosaics of change among regions and sites) but not in all respects. For example, there is a positive relationship between richness and Bray–Curtis turnover for all European biomes and for three out of the five biomes in North America (Extended Data Fig. 4a).

Even when trends are qualitatively similar, we observe differences in the timings of events. For instance, the different temporal trends in the 'same biome' type in Europe and North America (for example, Fig. 3i,m) highlight their different human and land management histories. There is debate about the duration and intensity of pre-Columbian human impact in the Americas[40–42] but our biome-specific analyses show turnover increasing since ~3,000 cal yr BP

in all but one of the North American biomes, whereas European turnover began to increase from ~9,000 cal yr BP depending on the biome considered (Fig. 3). Despite these different timings, relationships between ALCC and diversity metrics between the 'same' forested biome types in North America and Europe appear reasonably consistent (Fig. 5). In Eurasia, the case for an 'early Anthropocene' driven by the relatively early onset of wide-scale human impacts on the environment and climates has been strongly made[43,44], with the frequency and magnitudes of landscape and vegetation modifications increasing since the Mid Holocene and accelerating since the Late Bronze Age[33]. European biomes with the earliest signals of deforestation (the temperate forested and Mediterranean zones) exhibit the earliest signs of increased turnover (Fig. 3i,m,u) in tandem with rapid richness increases (Fig. 3j,n,v). They also have the strongest positive relationships of turnover and diversity increases with increasing levels of human land-cover change (Fig. 5).

In conclusion, the last four millennia can be characterized by an increasing human impact on global environments and biotas but the specific nature of those impacts varies geographically and through time (Figs. 2 and 3 and Extended Data Figs. 3, 5 and 6). Equally, their relationships with human land use also vary (Figs. 4 and 5 and Extended Data Fig. 7). This highlights the complexity of developing conservation strategies, given that the individual human histories and biodiversity trends of sites, biomes and continents vary and exclusion of historical human management and other influences may risk reducing rather than increasing local diversity in many instances[45].

## Methods

### Data collection

We downloaded all available pollen and associated chronological data from the Neotoma Paleoecology Database[46] (record $n$ = 6,097, database accessed November 2023) using the 'neotoma2' package[47]. Each record represents several samples from different deposit depths and their corresponding estimated ages.

We performed all data manipulations and analyses in this study in R[48] and all packages mentioned throughout are software extensions to R.

### Data filter

Pollen records held on Neotoma are compiled from a range of studies that use varying methods relating to pollen and chronological control sampling. Given the heterogeneous sampling effort between studies, we filtered pollen records on the basis of (1) the dating intensity of the chronological controls of a record and (2) the number of pollen samples per unit time. If a pollen record satisfied criteria relating to both its chronology and its pollen sampling, it was included in this study.

### Data quality inclusion criteria

We included pollen records in our analyses if (1) there were three or more Holocene chronological controls (that is, dated samples, usually radiocarbon dates) present along the record to ensure robust sample age–depth models, (2) there were no more than 3,000 years between any two chronological control points, for the same reason and (3) there were at least five Holocene pollen samples along the pollen record, each of which included at least 300 terrestrial pollen grains (excluding spores), to ensure that pollen-type diversity (for a standardized number of pollen grains, below) could be compared within and between records, through time. We analyse pollen abundance data from samples with ages younger than 11,700 cal yr BP and older than 100 cal yr BP (to avoid surface pollen samples being included in the dataset). If only specific sections of some pollen records met these criteria, we included these sections only and rejected sections along the same record that did not meet our inclusion criteria. If more than one section of a record met these criteria, but the record as a whole did not, we included the section with the most chronological control points.

The result of this filter was to reduce the initially downloaded 6,097 pollen records down to 1,763 Holocene (11,700–100 cal yr BP) pollen records (representing 64,305 samples). From the total pool of 1,763 records, across all 1,000 resamples (discussed below), the median number of pollen records included in any single analysis was 1,747 pollen records (minimum, 1,740; maximum, 1,752). On average, each record used in the analysis was represented by ~36 temporally separated samples (see Extended Data Fig. 2 for descriptions of the pollen and chronological data used).

### Pollen harmonization

To ensure that our pollen diversity results were comparable between pollen records, we harmonized the taxonomies and nomenclature of all pollen names present across the 1,763 pollen records included in these analyses. To do this, we used the harmonization tables provided in ref. 49 for all regions except Africa, for which we used the harmonization table published by ref. 15.

### Pollen resampling procedure

This study investigates the compositional turnover rate, richness, evenness and spatial heterogeneity of pollen samples through the Holocene and their relationships with human-induced land-cover changes. As in all ecological studies that sample individuals from a population, the greater the number of individuals (pollen grains) sampled from a population (sediment subsample), the more likely rare types will be identified. Given the methodological and geographical variation across the 1,763 pollen records, the number of pollen grains identified per sample varies among records (from our filtered minimum of 300 grains up to many thousands of grains in one sample). This range has potential implications for all six diversity metrics included in this study.

To account for this variation in pollen counts between samples, we randomly resampled 300 pollen grains from each pollen sample (which necessarily have at least 300 pollen grains due to our filtering procedure) without replacement, using the 'rrarefy' resampling function from the 'vegan' package[50]. This resampling results in a new global dataset with uniform pollen counts (300 pollen grains) for each sample. We repeated this resampling procedure 1,000 times to capture the variation inherent in this procedure, generating 1,000 global pollen datasets, each with 300 pollen grains per sample.

This work calculates six measures of pollen diversity through time, with each measure potentially affected differently by our resampling procedure. Clearly, increasingly stringent inclusion criteria reduce the total number of samples and records available for these analyses, yet overly downsampling pollen samples risks diversity estimates that are not robust to issues relating to varying pollen production, dispersal, preservation and so on. Resampling pollen samples down to a uniform 300–500 grains is suggested in the literature as sufficient to obtain robust diversity estimates[51]. We choose the lower end of this suggested limit, resampling all pollen samples down to 300 grains as a trade-off between data inclusion and robust diversity estimates. We have not performed a full adjustment of pollen counts, accounting for the relative pollen production of individual taxa, given our spatially extensive focus and the lack of relative pollen production estimates, fall speeds and so on for all pollen types included in our analyses.

### Age–depth models

We constructed Bayesian age–depth models for pollen records that met our inclusion criteria using the package 'Bchron'[52]. We selected either the IntCal20 (ref. 53) or SHCal20 (ref. 54) radiocarbon calibration curve to calibrate radiocarbon dates depending on latitude (IntCal20 to calibrate radiocarbon dates located in the Northern Hemisphere and SHCal20 for radiocarbon dates located in the Southern Hemisphere; atmospheric carbon isotopic ratios vary over time and by latitude, so age estimates need to account for this). We did not use a mixed radiocarbon calibration curve for records in the tropics/subtropics,

given the long timescales involved (multimillennial) and the coarse nature of these analyses, following the recommendation of the SHCal20 authors[54].

We ran each age–depth model using the default priors with 50,000 iterations and a thin value of 40. We discarded the first 10,000 iterations and did not allow age extrapolation to depths beyond 3,000 years from the oldest/youngest date. Rather than taking a measure of central tendency from the posterior distributions of the Bchron age–depth models, for each pollen record, we assigned to it an individual draw (a Monte Carlo sample) from its corresponding age model distribution. We repeated this independently for each of the 1,000 resampled global pollen datasets, thereby incorporating in the analyses uncertainties inherent in age–depth modelling procedures[55].

There is a degree of uncertainty associated with the modelled age of any given pollen sample inferred by any age–depth model (which predicts a distribution of potential dates per sample, rather than a unitary value). Therefore, a pollen sample with an age close to either the oldest (11,700 cal yr BP) or most recent (100 cal yr BP) limits of our specified age range may fall in or out of the dataset depending on the particular draw from the age–depth model, which may have consequences for the inclusion of the pollen record (given the requirement for pollen records to contain five or more Holocene samples). As such, the realized sample sizes varied slightly among the 1,000 different resampled datasets. These exclusions resulted in slightly different sample sizes for each of the 1,000 resampled datasets, representing a <0.1% reduction (on average) in numbers of samples included (below the potential maximum of 63,420) and a <0.25% reduction (on average) in the number of records included (below the potential maximum of 1,747) per analysis (that is, per resample; see section on Data filter for further details). Similarly, for the heterogeneity analysis, each individual pollen sample may fall into different time bins (time bins discussed below) depending on the particular draw from an age model. As such, our resampling procedure incorporates the overall age probability distribution for each pollen sample.

### Raw diversity calculations
For the within-record (that is, within site) diversity analyses (compositional turnover, richness and evenness), we first calculated the diversity measures and then partitioned these results into continental (see Fig. 1 for continental groupings of pollen records) and biome-within-continent diversity sets (using the World Wildlife Fund terrestrial ecoregions of the world biome categorization as our biome limits[19]). For the between-record analysis (spatial heterogeneity), we first partitioned the pollen records into continent and biome-within-continent subdatasets and then calculated the spatial heterogeneity of the pollen samples within each subdataset. We analysed the pollen data at the biome scale for Europe and North America only, given the temporal and spatial density of data in these regions. The following provides an overview of the methods used to calculate each diversity measure, with further detail contained in Supplementary Note 1.

### Compositional turnover rate
We quantified the turnover of pollen-type abundances through a pollen record using the Bray–Curtis index of dissimilarity[56], computed using the 'vegdist' function from the package 'vegan'[50]. We also quantified the turnover of pollen-type identities through a pollen record using the incidence-based Jaccard index[57], computed using the 'beta.pair' function from the package 'betapart'[58]. For both measures, the greater the dissimilarity between any two successive pollen samples, the greater the difference in their compositions and the greater the turnover of pollen types between them.

Pollen samples separated by a greater time interval are expected to have experienced a greater degree of turnover than those separated by a shorter interval. This needed to be accounted for in the analysis because the time intervals between consecutive samples vary within and between records. For both turnover measures, we estimated the relationship between turnover and the time interval between all pollen samples. Then for each pair of pollen samples, we subtracted the expected turnover from the raw turnover value, which resulted in a turnover value adjusted for the interval between the two pollen samples (Supplementary Note 1).

### Richness
Richness is a measure of the number of taxa (here pollen types; see below) present in a sample analysed with the same sampling effort, that is, a rarefied dataset, and is a widely used measure of diversity across both modern and palaeo-ecological studies. Although the relationship between pollen richness and the richness of the plant communities the pollen represents has been debated in the literature, many studies demonstrate a positive relationship between them[25,27,28,59,60] but this is not always the case (for example, refs. 61,62) (see Supplementary Note 1 for further discussion relating to pollen richness estimates).

### Evenness
Pielou's evenness[63] is a measure of inequality, identifying the distribution of abundances between pollen types in samples (see Supplementary Note 1 for detailed description). A low Pielou's evenness indicates a pollen assemblage dominated by a few pollen types, whereas a higher Pielou's evenness indicates a pollen assemblage more evenly represented by each pollen type. Communities with a higher evenness are therefore considered to have a greater diversity (see Supplementary Note 1 for discussion of the interplay between pollen richness and evenness).

### Heterogeneity
We quantified the compositional dissimilarity between pollen records (spatial beta diversity) at the continental scale and for the well-represented biomes in North America and Europe, in 500 year time bins, to understand whether vegetative communities were becoming more or less homogenous over the Holocene. We computed multisite extensions of the abundance-based Bray–Curtis measure and the incidence-based Jaccard measure to investigate beta diversity trends using the 'betapart' package[58]. We applied the same data and region filters to both the continental and the biome-within-continent heterogeneity analyses, only running the analysis on regions with 20 or more pollen records (see Supplementary Note 1 for full procedure).

### Modelling continental and biome-within-continent diversity trends
We model our empirical continental- and biome-scale temporal turnover, richness and evenness data with GAMs. GAMs are used to estimate smooth (nonlinear) relationships between predictor and response variables and are advocated in the literature for modelling palaeo-ecological data through time[64] (that is, time series). We model continental- and biome-scale diversity against time separately using two variations of the same model.

To assist with model convergence, for each diversity metric and each of the 1,000 resampled datasets, we standardized the diversity data within-continent (global analysis) or within-biome (within-continent analysis, separately for Europe and North America), subtracting from each diversity datapoint the mean diversity value computed across the entire continent or biome and dividing this centred value by the standard deviation of the continent or biome. We then fit a GAM to these standardized diversity values (Diversity_st, below) using the 'mgcv'[65] 'bam' function, estimating a smooth function of each diversity metric against time for each continent or biome. Using region to mean continent or biome as appropriate, each GAM took the form,

Diversity_st ~ region + s(age, by = region, bs = 'fs', k=50) + s(datasetid, bs = 're'), method = 'fREML', discrete = TRUE, family = family

Our continent and biome models follow the same specification, with 's' representing a 'smooth' function (created by a sum of 'k' basis functions) of the explanatory variable 'age' for each continent/biome. Given the variation in average diversity between pollen records, we include a random effect for the intercept term for each (s(datasetid, bs = 're')). We estimate a separate smooth for each continent/biome (s(age, by = region)), with k (determining the maximum complexity, or 'wiggliness', of the smooth function) set to 50 to allow for flexibility. These models allow the wiggliness, shape and intercepts of each smooth to vary by continent/biome, with a 'smoothing penalty' (estimated using fast restricted maximum likelihood, fREML) preventing overly wiggly smooths (that is, overfitting). Given the relatively large size of each dataset being modelled (~60,000 rows) and the many individual models being fit, we specify discrete = TRUE to speed up computation time and reduce memory usage (when comparing with and without discretization, differences in output were unidentifiable; see refs. 66,67 for details). As the conditional standardized diversity metrics generally exhibited heavy tails, we mostly used the scaled t family of distributions for our GAMs (family = scat()`), specifying a Gaussian distribution (family = gaussian()`) in cases where the conditional data were not heavy tailed.

After fitting each GAM, we applied the inverse of the standardization operation so that predictions are represented on the natural scale. Overall, our modelling choices detailed above reflect a combination of factors including the large number of GAMs being estimated (four diversity metrics in each resample (1,000)), our goals (estimating overall trends over multimillennial timescales) and the need to manage computational time (see refs. 64,68 for discussion of GAMs in (palaeo) ecological contexts or ref. 69 for detailed information relating to GAMs more generally).

We investigated the effect of including continental smooths on model fit by comparing the AIC of a model with a smooth for each continent included to the AIC of a model fit to the same data without continental smooths. Across most of the 1,000 turnover, richness and evenness datasets there was a reduction of two or more AIC units (that is, a significant improvement) when including continental smooths in the model (Supplementary Table 2). We repeated this procedure for the European and North American biome models, comparing whether including a smooth term for each biome improved the model fit compared with a model (including only a single, 'global' smooth) fit to the data without. Again, most richness, evenness and turnover models were improved (indicated by a reduction in AIC of two or more) after including a smooth for each biome (Supplementary Table 3) and we therefore included these terms.

Finally, we present the median *P* values for each smooth component (relating to continents/biomes) and diversity metric (across all 1,000 resamples; Supplementary Tables 1 and 4), with significant smooths suggesting a 'real' nonlinear effect of time (see ref. 70 for a description of the significance tests).

**Investigating effects of humans on observed trends**
We investigate the effects of human-generated land-cover change on the continental and biome diversity trends presented in Figs. 2 and 3 and Extended Data Fig. 3 using the global KK10 scenario of ALCC[16,17] as a human land-use variable. The KK10 estimates the total fraction of each 5 arcmin grid cell that is under any human land use over the period 8,000–100 cal yr BP in annual time slices. Briefly, the KK10 simulation is based on maps of present-day agriculture/pasture suitability, past population estimates and per capita land-use estimates[17]. Population estimates were compiled by the KK10 authors from the literature for the more recent past, using the estimates of ref. 71 as a starting point and from the Global Land Use and Technological Evolution Simulator[72,73] for the period 8,000–3,000 cal yr BP. The KK10 population density and ALCC relationship was initially developed using European data. However, the potential area for cultivation/grazing is much higher in

the tropics and much lower in the high latitudes. Therefore, the KK10 authors added a scaling term to the model to account for latitudinal variation in net primary productivity (see ref. 16 for further details).

In comparison to other ALCC simulations, for example HYDE[74,75], the KK10 scenario assumes a nonlinear relationship between human per capita land-use area and time, with per capita land use reducing nonlinearly over time, rather than remaining constant. Driven by increasing human populations, technological changes and associated land-use intensification, this negative nonlinear relationship has been shown to be a more accurate characterization than a linear one[76]. A European comparison of ALCC estimates of HYDE and KK10 with pollen-inferred deforestation patterns shows a stronger agreement with KK10 than HYDE[77].

The KK10 scenario provides an ALCC estimate for each year from 8,000 to 100 cal yr BP at a 5 arcmin spatial resolution. However, (1) the ALCC estimates are based on historical population reconstructions and agricultural/pastoral suitability maps for region × time combinations, rather than specific reconstructions for the precise landscape surrounding each individual pollen record and (2) our diversity estimates are summarized at continental and biome scales. Hence, we conducted this analysis at the regional (continental and biome-within-continent) level. For each year from 8,000 to 100 cal yr BP, we grouped the KK10 ALCC estimates by continent and biome-within-continent for Europe and North America and computed the mean value across all grid cells within each region. This provided a time series of ALCC values for each region, which we carried forwards to the subsequent modelling stage.

For each region (continent and biome-within-continents in Europe and North America), we modelled the relationship between each diversity metric (Diversity) and square-root transformed ALCC (sqrt_ALCC) in generalized linear mixed effects models using the 'bam' function from 'mgcv'[65]. To assist with model convergence, we standardized the turnover, evenness and heterogeneity values before running the models, performing the inverse of the standardization operation on the predictions to return them to the natural scale. We ran all regional models separately, each of which took the form, Diversity ~ sqrt_ALCC + s(datasetid, bs = 're').

We included a random intercept term for each pollen record given the variation in average diversity among records (s(datasetid, bs = 're')). The above model assumes that the dependent variable values (Diversity) are independent. However, given that the dependent variables are time series, this assumption is potentially violated via the autocorrelation of the Diversity values. To account for this, we assessed the autocorrelation of the above model using the 'acf' function in 'R'. If the calculated autocorrelations exceeded white noise 95% confidence intervals, we interpreted this as evidence for significant autocorrelation. In this case, we reran the above model but including the autocorrelation value at a lag of one using the rho argument in 'bam'. We plotted the residuals from models run on a random subset of the 1,000 resampled pollen datasets as a visual check and there were no patterns present. We present the model fits for each continent and biome-within-continent across the 1,000 resamples as medians and interquartile ranges in Figs. 4 and 5 and Extended Data Fig. 7.

**Variation in diversity changes among sites**
For three 1,850 year periods in the Holocene (the Early Holocene (11,700–9,851 cal yr BP), the Mid Holocene (6,725–4876 cal yr BP) and the Common Era (1–1850 CE)), we regressed each within-record diversity metric (turnover, richness and evenness) against time for each pollen record in linear models to assess whether individual pollen records showed broadly the same linear temporal trends or whether they varied (for example, whether some sites became richer while others became less rich).

Pollen richness values are integers bound between 1 and 300 (given our resampling; with medians up to the low 20s) and we therefore log transformed the richness values and fit a linear model of the form

log_richness ~ age using the 'lm' function. Evenness is a [0,1] bound variable, so we logit transformed these values and fit a linear model of the form logit_evenness ~ age using the 'lm' function. Likewise, compositional turnover is a [0,1] bound variable, so we logit transformed these values and fit a linear model of the form logit_turnover ~ age also using 'lm'. We extracted the slope coefficients from each model and multiplied them by 1,000 to provide an estimate of the millennial linear change in each metric within an individual pollen record (on the log and logit scale for richness and evenness/turnover, respectively), on the basis of the subset of data from each of the three periods. We present the slope coefficients from these linear models as boxplots (median, the interquartile range represented by the two outermost hinges of the boxplot and the whiskers representing 1.5 times the interquartile range). We present slope coefficients from linear models fit to pollen diversity data from pollen records with five or more samples. Continental results are shown in Extended Data Fig. 5 and North American and European biome results across all periods in Extended Data Fig. 6.

#### Depositional environment sensitivity analysis

Fossil pollen records archived on Neotoma are associated with sedimentary profiles extracted from a wide variety of depositional environments (that is, environmental settings), including lakes, mires, bogs, fens, forest hollows and so on. The depositional environment of a pollen record has the capacity to influence sampled pollen assemblages through factors relating to pollen source area and taphonomy, among others[18]. Of particular relevance to macroscale pollen diversity studies is pollen source area (the spatial area relating to pollen spectra preserved in a pollen record). Pollen records extracted from depositional environments associated with large pollen source areas (for example, large lakes) are, by definition, more likely to represent plant communities from a wider spatial area than those from smaller source areas (for example, peat bogs and forest hollows)[78,79]. In the context of pollen diversity analyses such as these, variation in pollen source area between records effectively alters the sampling areas of pollen records associated with some depositional environments compared to others, with implications for diversity measures.

To examine the influence of individual depositional environments on our results, we performed exploratory leave-one-out sensitivity analyses, by reproducing our analyses on each subset of the data with all pollen records from a single depositional environment excluded iteratively. If the modelled pollen diversity trends in Figs. 2–5 were driven primarily by any single pollen depositional environment or variation in the sample sizes of pollen records from each depositional environment, then predictions from models fit to the pollen dataset with that depositional environment left out would strongly differ from predictions made from models with that depositional environment still included and also from the results contained in Figs. 2–5 (which are based on all pollen records). A total of 82 unique depositional environments are represented in our dataset of 1,763 pollen records. We therefore reran the analyses behind Figs. 2–5 with pollen records associated with each depositional environment left out iteratively (resulting in up to 82 runs per region). Given the computational expense involved with rerunning our analyses 82 times, we restrict our analyses to the median diversity values across all 1,000 resamples, recalculated for each subset.

We present results from our sensitivity analyses in Extended Data Fig. 8 for the diversity through time analyses and Extended Data Figs. 9 and 10 for the analyses relating pollen diversity metrics to ALCC. Across all regions considered, the functional forms of each of the 82 reruns are generally qualitatively similar both to the other runs (per region) and also to the results in Figs. 2–5. Intercepts do vary somewhat between runs, although this is expected given that average richness is likely to be related to pollen source area. There is also generally more variation among runs for regions with lower sample sizes. Removing pollen records from any depositional environment results in too few records to meet our criteria for analysing Australasian pollen heterogeneity.

Overall, these results indicate that our main results are not driven by patterns from any single depositional environment or by changing frequencies of pollen records from different depositional environments through time.

#### Reporting summary

Further information on research design is available in the Nature Portfolio Reporting Summary linked to this article.

### Data availability

All pollen, chronological and human land-use data used in this study are open access and their locations detailed in the main text or Methods. Supplementary Table 5 details the pollen records included in these analyses.

### Code availability

Analysis code is archived on Zenodo and is available at https://doi.org/10.5281/zenodo.11395089 (ref. 80).

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

## Acknowledgements

Data were obtained from the Neotoma Paleoecological Database (http://www.neotomadb.org) and its constituent databases (the African Pollen Database, the Indo-Pacific Pollen Database, the European Pollen Database and the Alpine Pollen Database, the Latin American Pollen Database and the North American Pollen Database). The work of data contributors, data stewards and the Neotoma community is gratefully acknowledged. We thank C. Beale and J. Hatfield for advice on the computation and numerical analyses and M. Stratigos for helpful discussions relating to radiocarbon dating. We also thank C. Li for assistance with analyses relating to previous versions of this paper. We thank T. Giesecke, J. Hatfield, I. Lazagabaster, C. Lyon, M. Stratigos and K. Walsh for comments on versions of the paper. This work was funded by a Leverhulme Trust Research Centre—The Leverhulme Centre for Anthropocene Biodiversity, grant no. RC-2018-021 (J.D.G., B.F., N.M. and C.D.T. all recipients). The Viking cluster was used during this project, which is a high-performance compute facility provided by the University of York. We are grateful for computational support from the University of York, IT Services and the Research IT team. Correspondence and requests for materials should be addressed to J.D.G. For the purpose of open access a Creative Commons Attribution (CC BY) licence is applied to any Author Accepted Manuscript version arising from this submission.

## Author contributions

J.D.G., N.M., B.F. and C.D.T. developed and contributed to the design of analyses. J.D.G. and B.F. carried out analyses. J.D.G. and C.D.T. wrote the first draft of the paper and all authors commented on it.

## Competing interests

The authors declare no competing interests.

## Additional information

**Extended data** is available for this paper at https://doi.org/10.1038/s41559-024-02457-x.

**Correspondence and requests for materials** should be addressed to Jonathan D. Gordon.

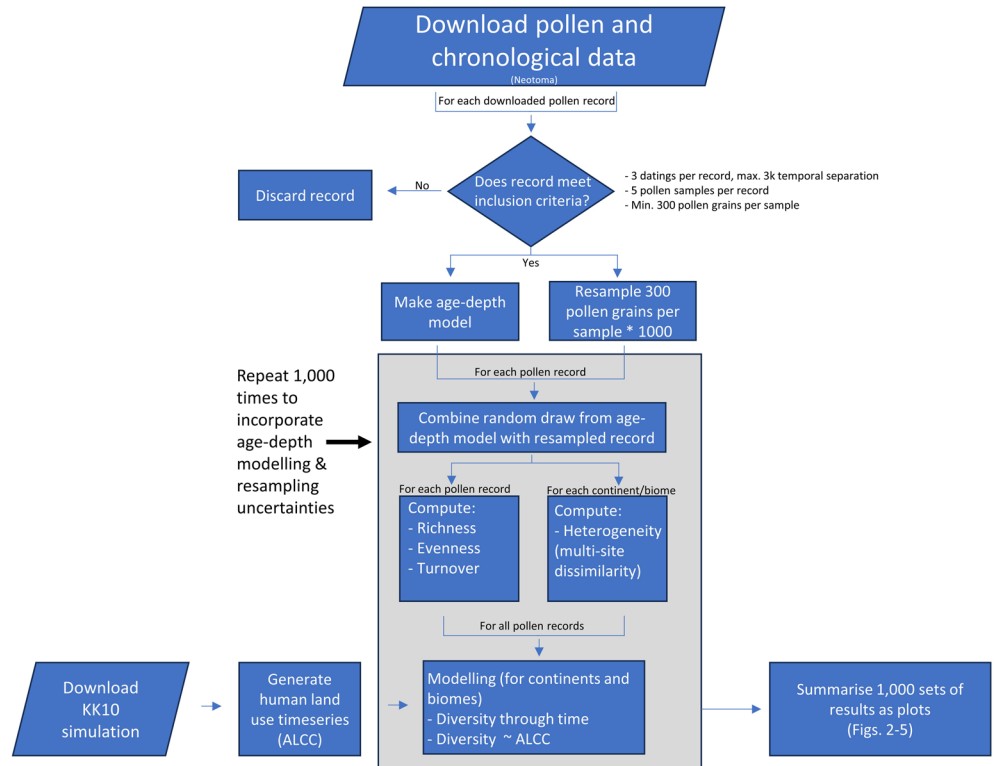

**Extended Data Fig. 1 | Methodological overview.** Figure highlights the main stages of these analyses, with key decision points described.

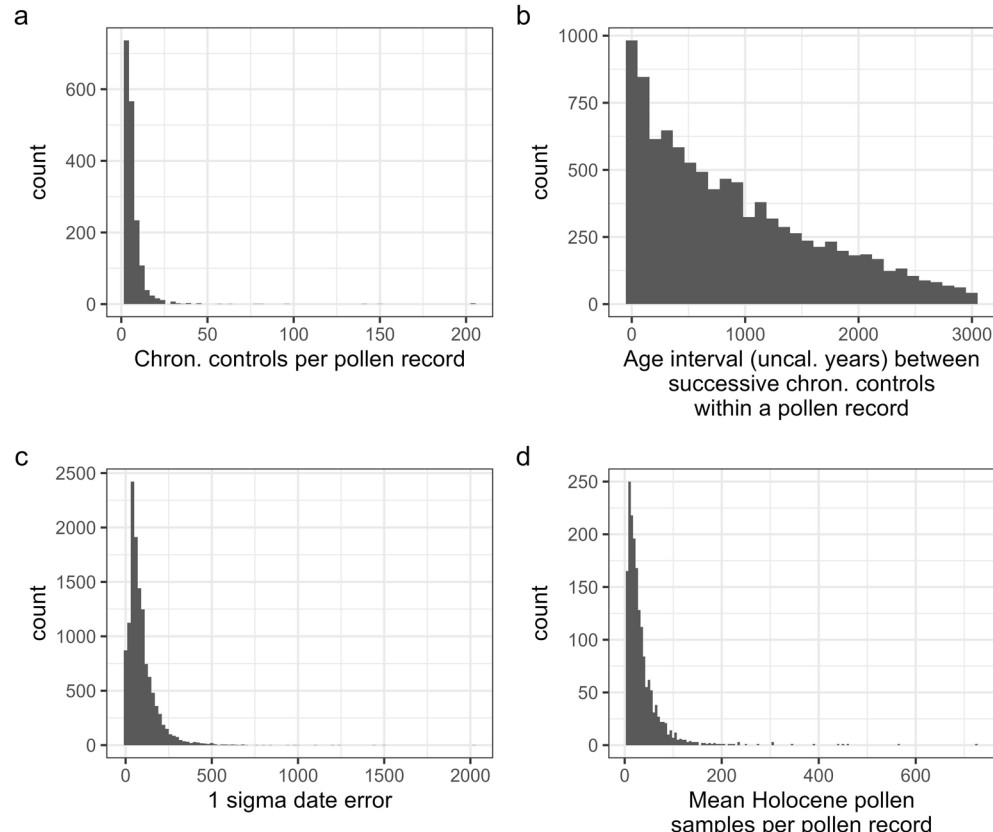

**Extended Data Fig. 2 | Pollen and dating sampling intensity. a**) Number of chronological control points associated with each pollen record; **b**) median age intervals between successive chronological controls along a pollen record; **c**) date uncertainties (1 standard deviation) of chronological controls; **d**) mean number of Holocene pollen samples per pollen record.

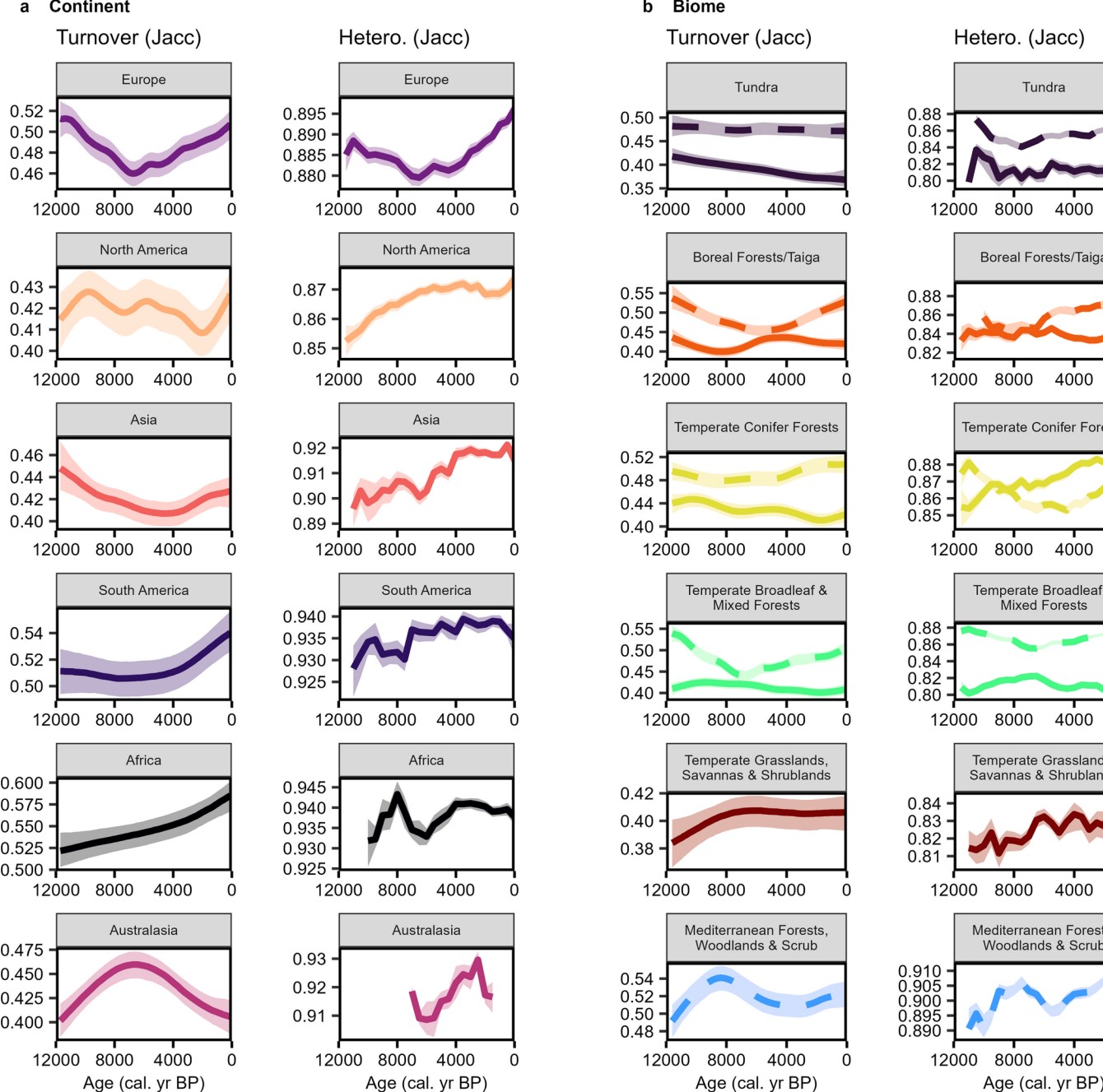

**Extended Data Fig. 3 | Temporal Holocene Jaccard turnover and heterogeneity patterns for (a) continents and (b) biomes within North America (solid lines) and Europe (dotted lines).** Turnover data (columns 1 & 3) are fit against time in GAMs. Heterogeneity data (column 2 & 4) are not fitted GAMs, but distributions of data (given the reduced heterogeneity sample size arising from the 500 year binning, see *Methods*). The 1,000 resamples are summarized by medians (thick lines) and interquartile ranges (shaded intervals). Sample sizes per region (continent/biome) are identical to those of the respective regions in Figs. 2 and 3.

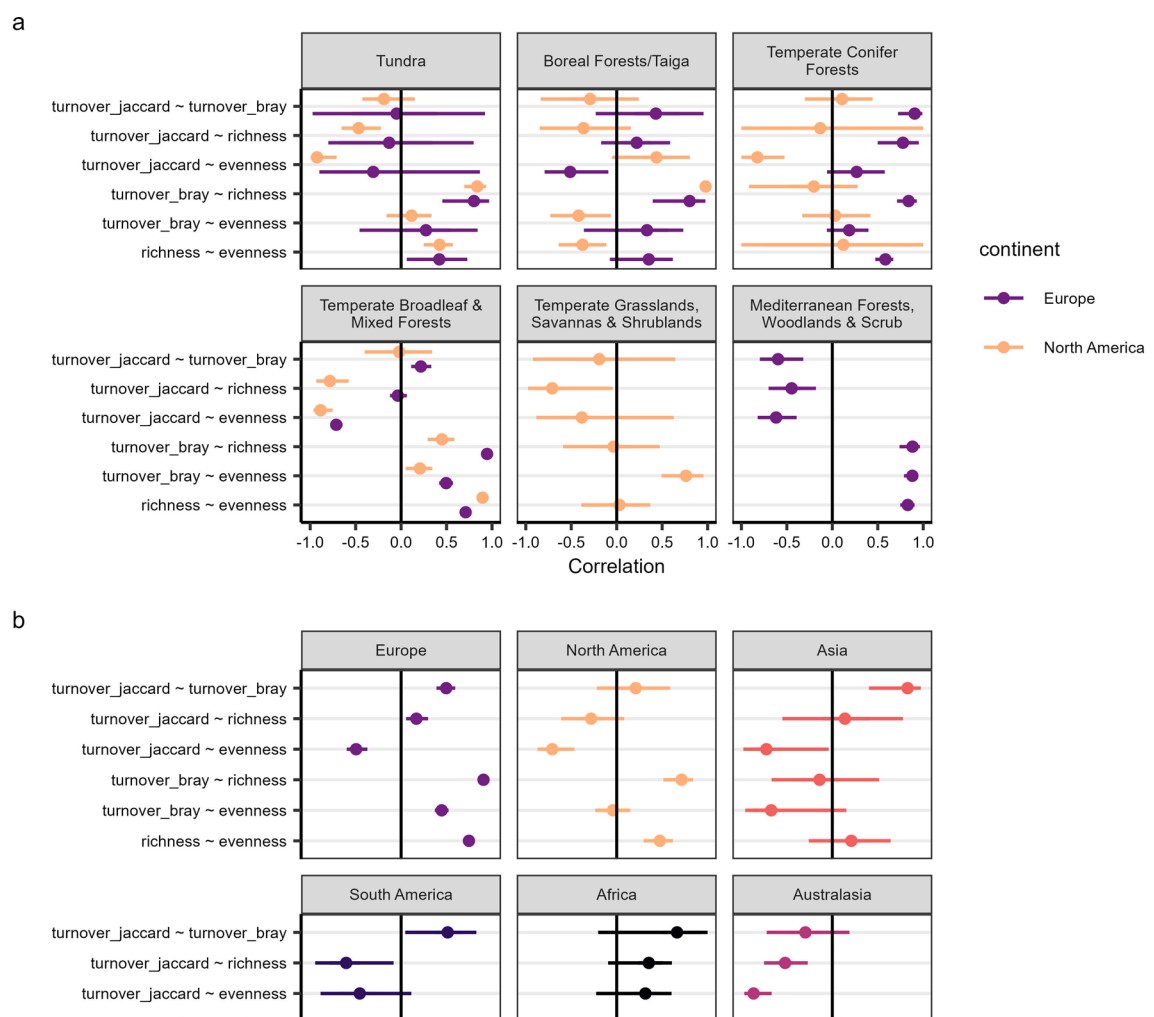

**Extended Data Fig. 4 | Pearson's product-moment correlations between GAMs fitted to diversity metrics for (a) continent and (b) biomes within North America and Europe.** Points represent the median correlation across the 1000 resamples, the thick and thin bars represent 50% and 95%, respectively, of the correlation values across all 1000 resamples. Note that heterogeneity is not included because we did not fit a GAM to these data.

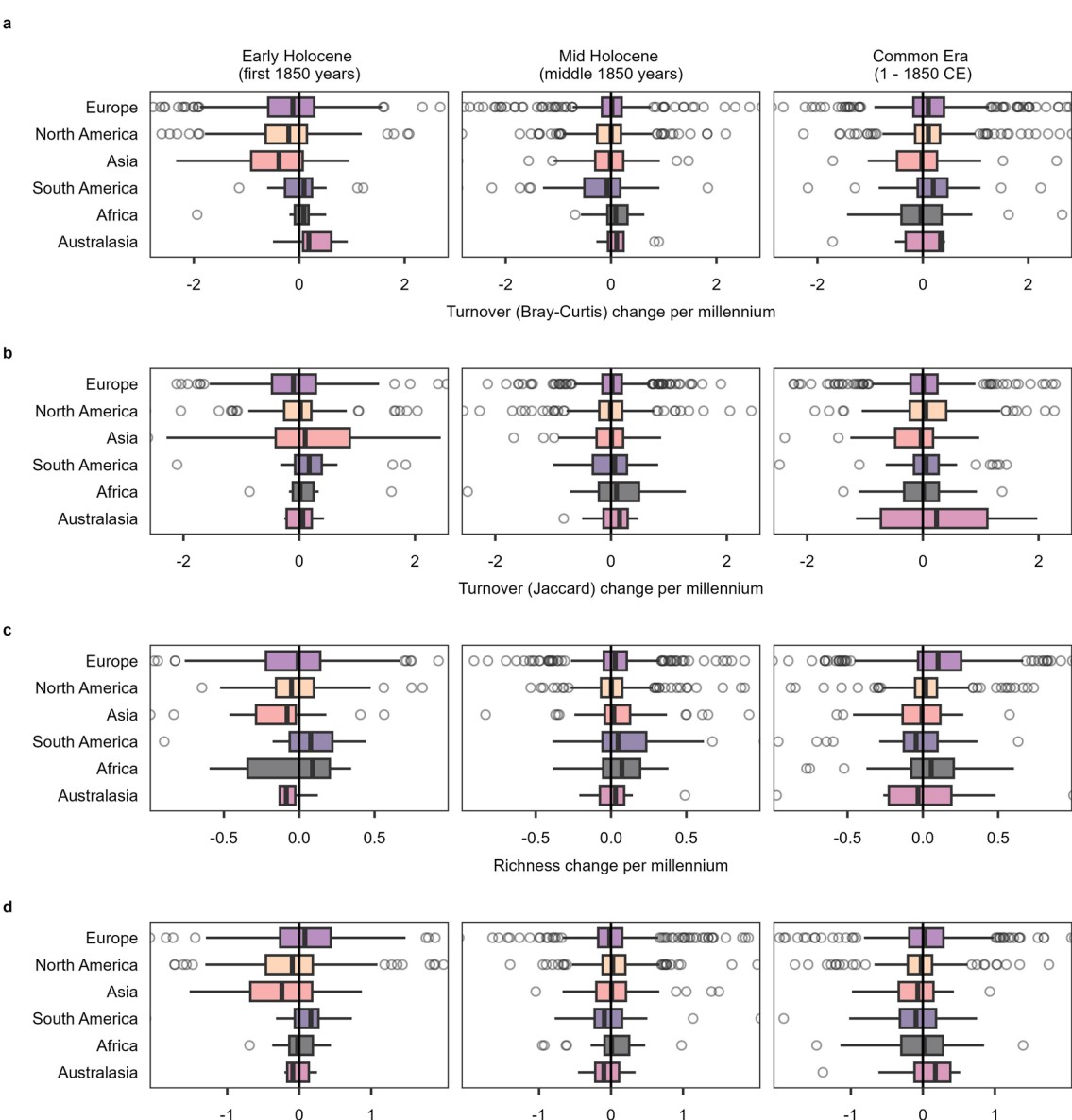

**Extended Data Fig. 5 | Site-level (pollen record) estimates of millennial changes in diversity over three 1850 year periods over the Holocene: The early Holocene, mid Holocene and the Common Era. a**) turnover (Bray-Curtis), **b**) turnover (Jaccard), **c**) richness, **d**) evenness. Values computed by regressing each diversity measure onto age (calibrated years BP), extracting the slope coefficients and multiplying them by 1000 to give the expected linear change in diversity per millennium (on the log and logit scale for richness and evenness/ turnover, respectively). Box and whiskers represent the cohort of pollen records from each continent. Plots are zoomed in for clarity of presentation, but retain 95% of slope estimates. In the following, 'EH' represents Early Holocene, 'MH' represents Mid Holocene and 'CE' represents Common Era: Africa EH $n$ = 111, MH $n$ = 334, CE $n$ = 669; Asia EH $n$ = 176, MH $n$ = 626, CE $n$ = 715; Australasia EH $n$ = 85, MH $n$ = 171, CE $n$ = 149; Europe EH $n$ = 3,864, MH $n$ = 11,320, CE $n$ = 9,827; North America EH $n$ = 2,476, MH $n$ = 4,514, CE $n$ = 4,123; South America EH $n$ = 198, MH $n$ = 676, CE $n$ = 908.

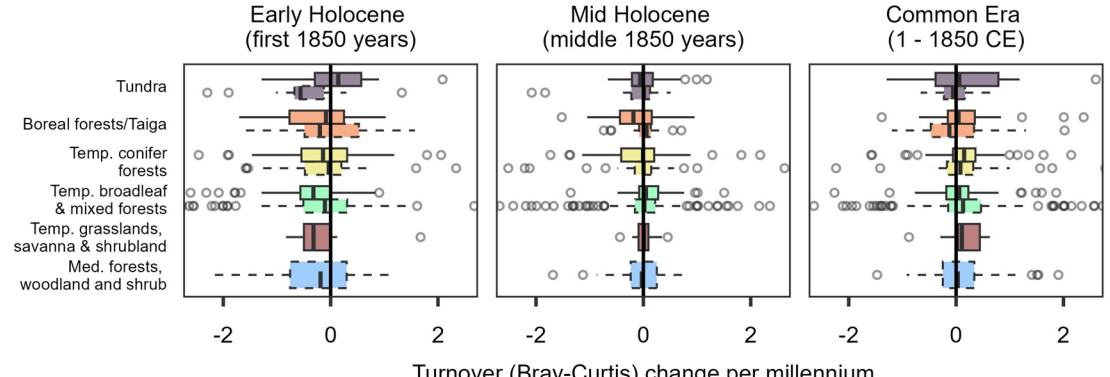

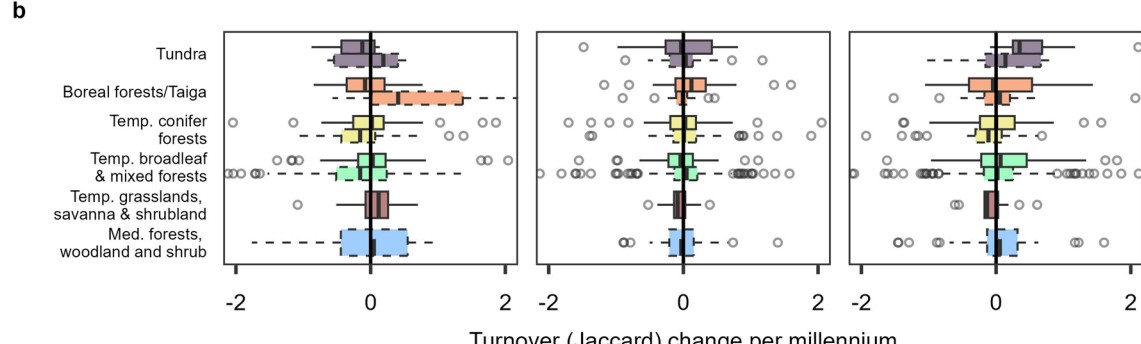

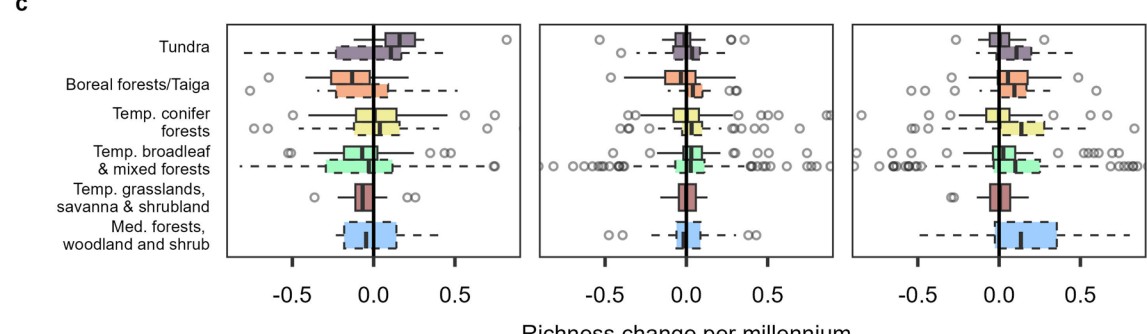

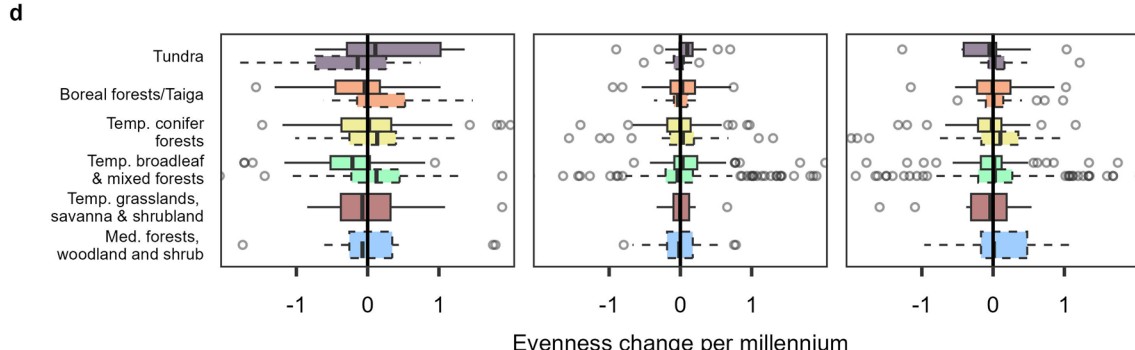

**Extended Data Fig. 6 | See next page for caption.**

**Extended Data Fig. 6 | , Site-level (pollen record) estimates of millennial changes in diversity over three 1850 year periods over the Holocene: The early Holocene, mid Holocene and the Common Era, by European and North American biome. a)** turnover (Bray-Curtis)**, b)** turnover (Jaccard), **c)** richness**, d)** evenness. Values computed for data-rich records (see *Methods*) by regressing each diversity measure onto age (calibrated years), extracting the slope coefficients and multiplying them by 1000 to give the expected linear change in diversity per millennium (on the log and logit scale for richness and evenness/turnover, respectively). Box and whiskers represent the cohort of pollen records from each biome. Biomes included depending on data inclusion criteria detailed in *Methods*. Box and whiskers represent the cohort of pollen records from each biome. Plots are zoomed in for clarity of presentation, but retain 95% of slope estimates. Solid boxes represent North American records, dashed boxes represent European records. In the following, 'NA' represents North America, 'EU' represents Europe, 'EH' represents Early Holocene, 'MH' represents Mid Holocene and 'CE' represents Common Era: NA Tundra, EH $n = 149$, MH $n = 355$, CE $n = 209$; EU Tundra, EH $n = 148$, MH $n = 493$, CE $n = 186$; NA Boreal forests/taiga, EH $n = 217$, MH $n = 895$, CE $n = 405$; EU Boreal forests/taiga, EH $n = 111$, MH $n = 633$, CE $n = 415$; NA temperate conifer forests, EH $n = 666$, MH $n = 1,139$, CE $n = 1,125$; EU temperate conifer forests, EH $n = 1,058$, MH $n = 1,637$, CE $n = 813$; NA temperate broadleaf and mixed forests, EH $n = 1,119$, MH $n = 1,669$, CE $n = 1,886$; EU temperate broadleaf and mixed forests, EH $n = 2,043$, MH $n = 7,373$, CE $n = 7,535$; NA temperate grasslands savannas and shrublands, EH $n = 254$, MH $n = 386$, CE $n = 349$; EU Mediterranean forests, woodland and scrub, EH $n = 499$, MH $n = 1,131$, CE $n = 878$.

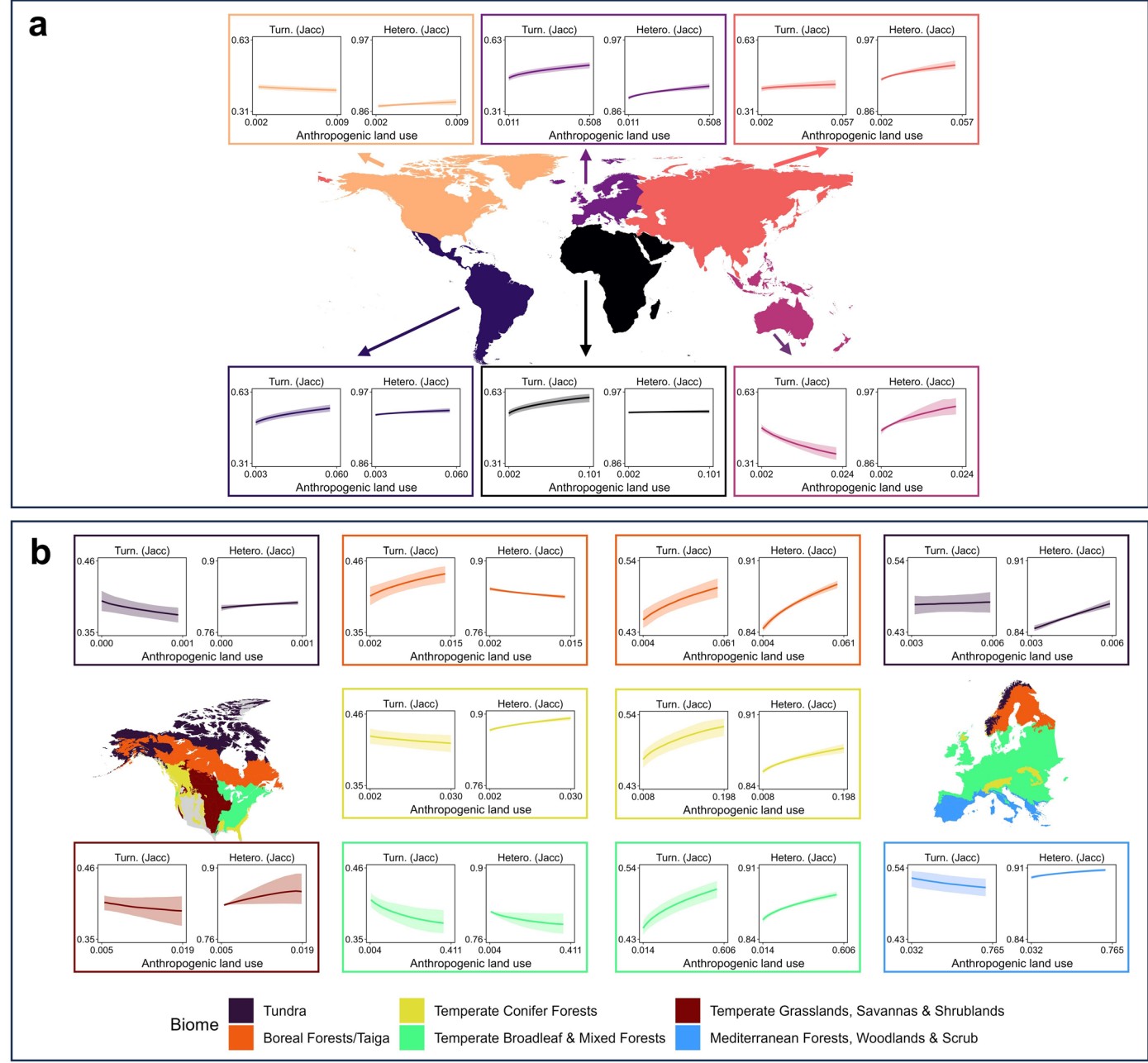

**Extended Data Fig. 7 | Fitted relationships between Jaccard diversity metrics and anthropogenic land use for (a) continents and (b) North American and European biomes.** Lines represent median fits across 1,000 resamples, shaded intervals represent interquartile ranges across 1,000 resamples. Sample sizes per region (continent/biome) are identical to those of the respective regions in Figs. 4 and 5.

**a**

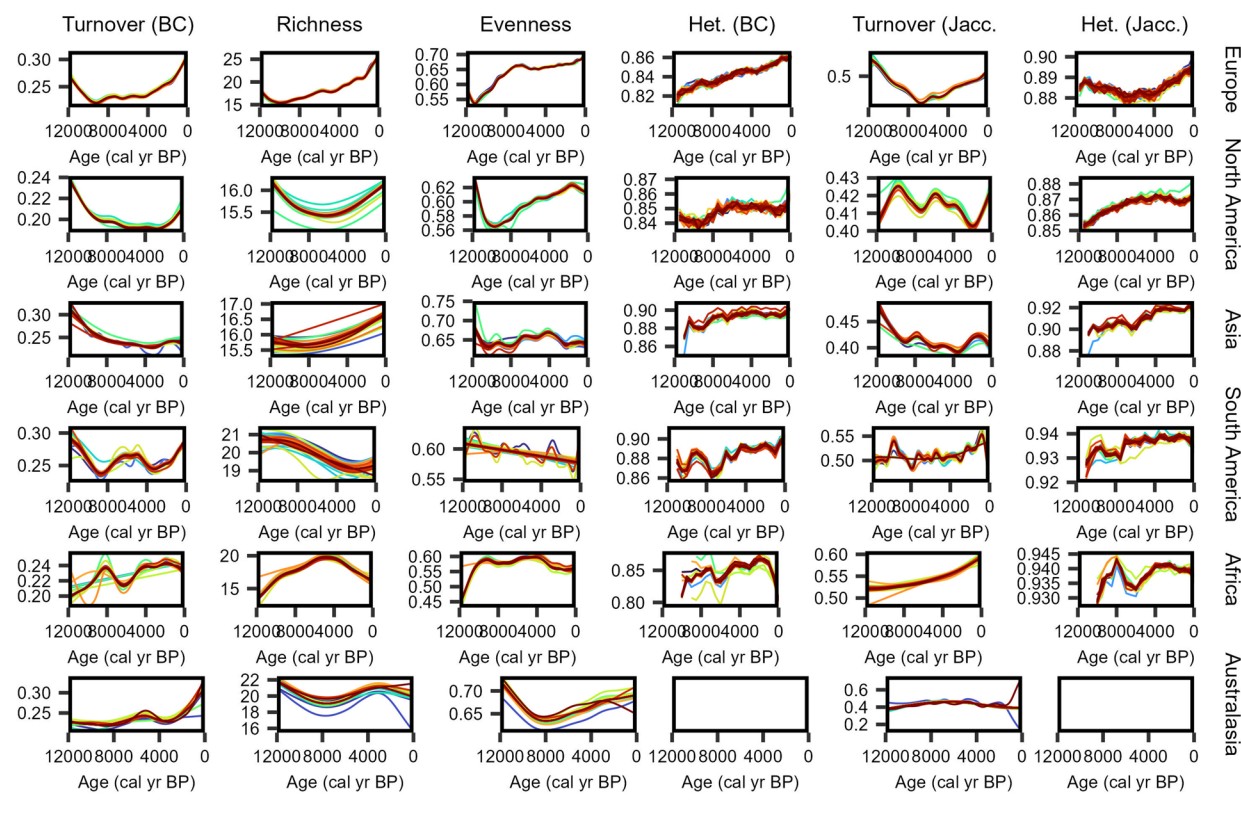

**b**

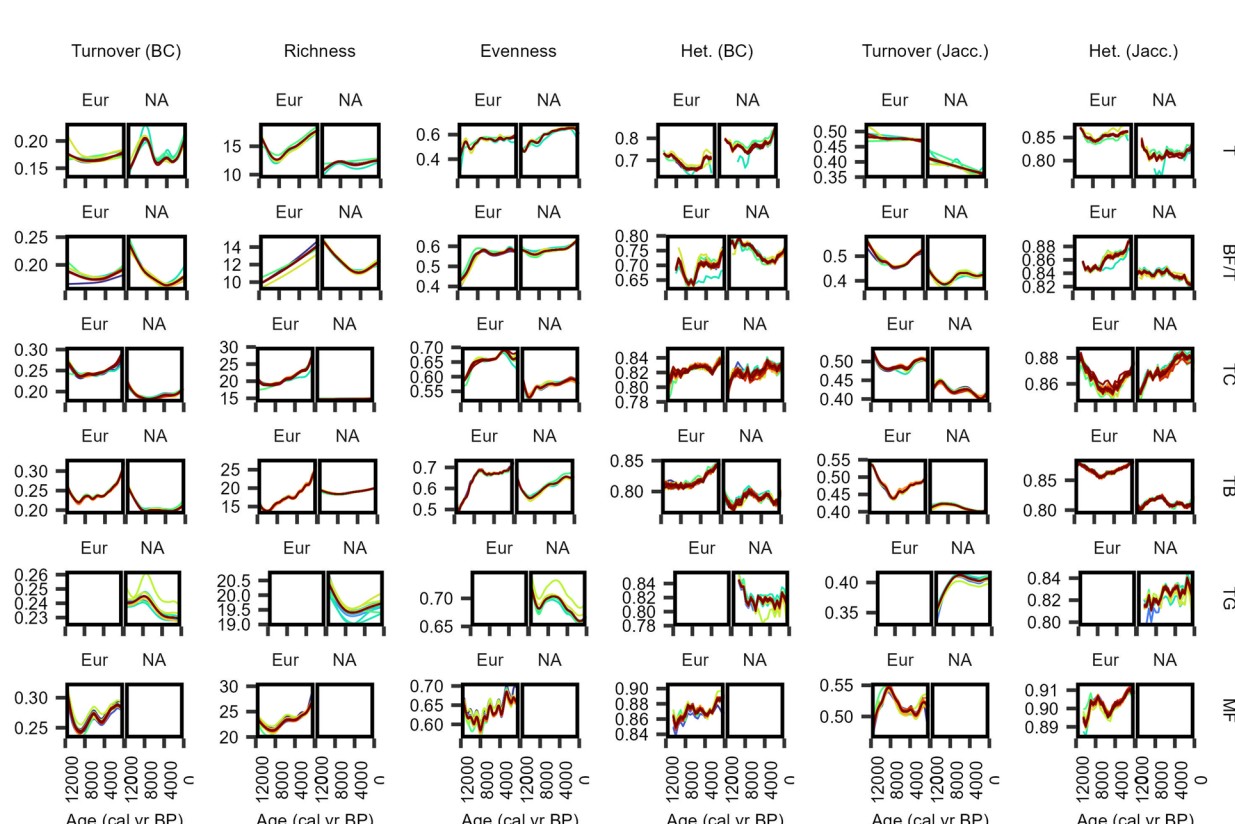

**Extended Data Fig. 8 | See next page for caption.**

**Extended Data Fig. 8 | Leave-one-out (LOO) sensitivity analysis for temporal (a) continental and (b) North American and European and biome diversity patterns.** Coloured lines represent GAMs fit to the diversity data minus all pollen records from each depositional environment iteratively removed (see *Methods*). Heterogeneity panels are not GAMs, but distributions of the 82 runs of LOO datasets. In panel (b), 'EU' = Europe, 'NA' = North America, 'T' = Tundra, 'BF/T' = Boreal forests/Taiga, 'TC' = Temperate conifer forests, 'TB' = Temperate broadleaf & mixed forests, 'TG' = Temperate grasslands, savannas & shrublands and 'MF' = Mediterranean forests, woodlands & scrub.

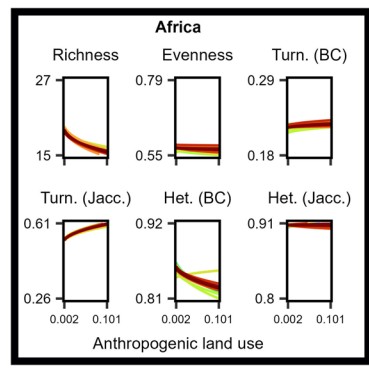
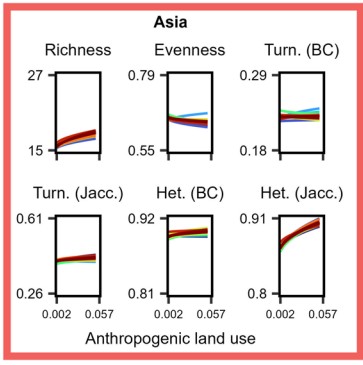
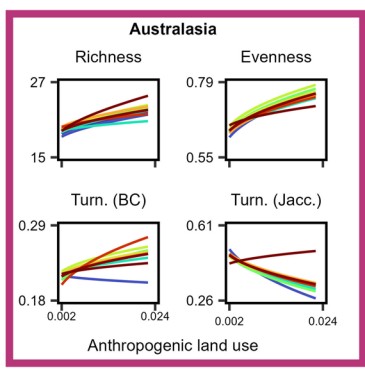
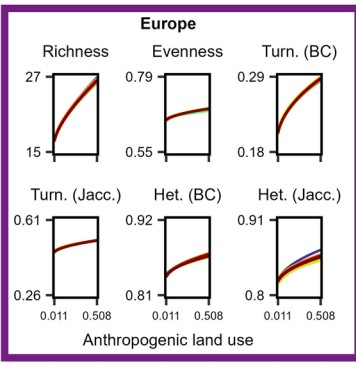
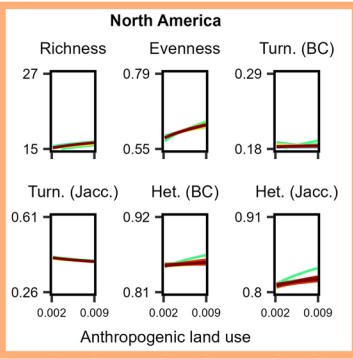
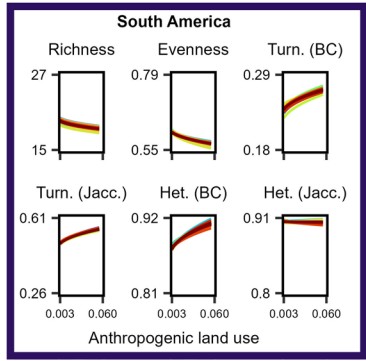

**Extended Data Fig. 9 | Leave-one-out (LOO) sensitivity analysis, fitted relationships between each diversity metric and anthropogenic land use data for each continent for data encompassing the period 8,000–100 cal BP.** Lines represent a fit to the pollen diversity data minus all pollen records from each depositional environment iteratively removed. Note that ALCC is back transformed and axis ranges differ between continents. LOO analysis for Australasian heterogeneity could not be performed due to small sample sizes (removal of a depositional environment resulted in the remaining data not passing the inclusion criteria for heterogeneity analysis, see *Methods*). 'Turn' = turnover, 'Het.' = heterogeneity, 'BC' = Bray-Curtis, 'Jacc.' = Jaccard. 'Turn. (Jacc)' therefore means turnover as calculated by the Jaccard dissimilarity index.

## North_America

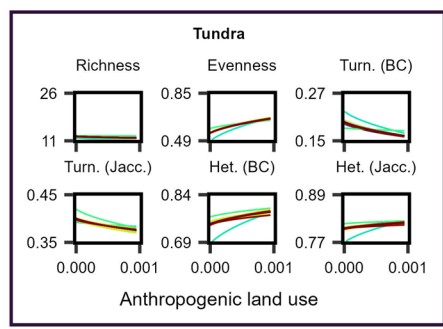
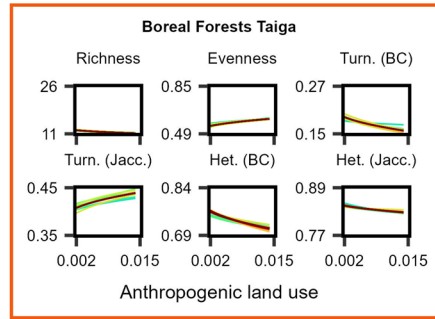
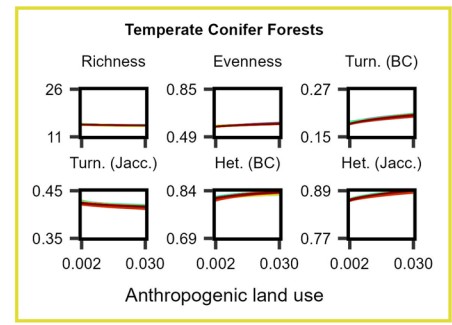

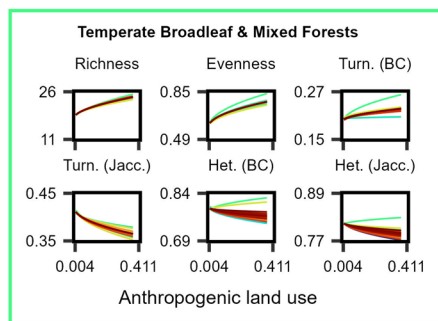
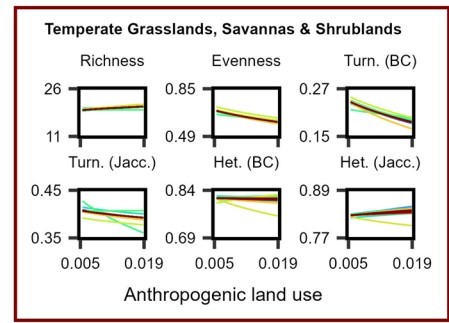

## Europe

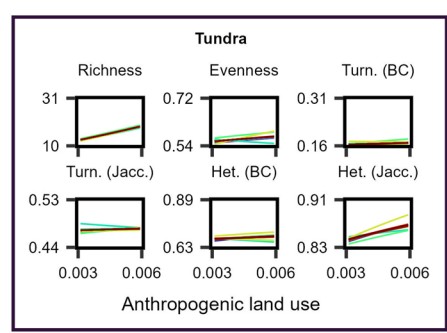
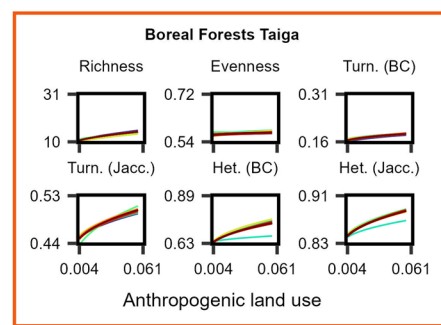
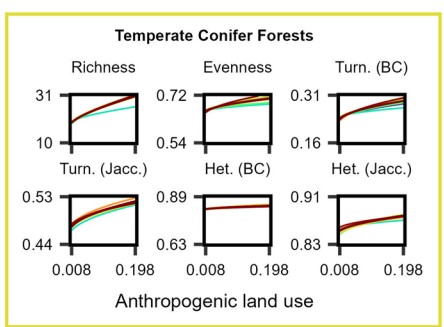

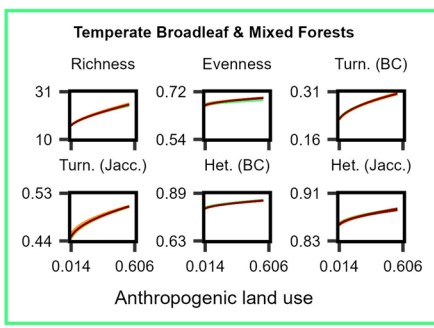
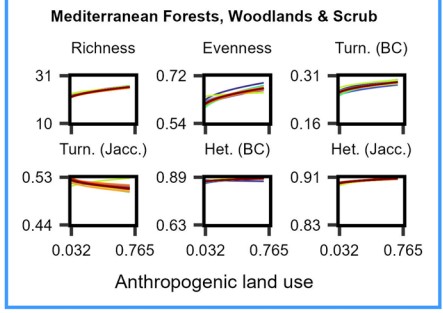

**Extended Data Fig. 10 | Leave-one-out sensitivity analysis, fitted relationships between each diversity metric and anthropogenic land use data for biomes in North America and Europe for data encompassing the period 8,000–100 cal BP.** Figure structure identical to Extended Data Fig. 9.

# Reporting Summary

## Statistics

For all statistical analyses, confirm that the following items are present in the figure legend, table legend, main text, or Methods section.

| n/a | Confirmed | |
|---|---|---|
| ☐ | ☒ | The exact sample size (*n*) for each experimental group/condition, given as a discrete number and unit of measurement |
| ☐ | ☒ | A statement on whether measurements were taken from distinct samples or whether the same sample was measured repeatedly |
| ☐ | ☒ | The statistical test(s) used AND whether they are one- or two-sided *Only common tests should be described solely by name; describe more complex techniques in the Methods section.* |
| ☐ | ☒ | A description of all covariates tested |
| ☒ | ☐ | A description of any assumptions or corrections, such as tests of normality and adjustment for multiple comparisons |
| ☐ | ☒ | A full description of the statistical parameters including central tendency (e.g. means) or other basic estimates (e.g. regression coefficient) AND variation (e.g. standard deviation) or associated estimates of uncertainty (e.g. confidence intervals) |
| ☒ | ☐ | For null hypothesis testing, the test statistic (e.g. *F*, *t*, *r*) with confidence intervals, effect sizes, degrees of freedom and *P* value noted *Give P values as exact values whenever suitable.* |
| ☐ | ☒ | For Bayesian analysis, information on the choice of priors and Markov chain Monte Carlo settings |
| ☐ | ☒ | For hierarchical and complex designs, identification of the appropriate level for tests and full reporting of outcomes |
| ☐ | ☒ | Estimates of effect sizes (e.g. Cohen's *d*, Pearson's *r*), indicating how they were calculated |

*Our web collection on statistics for biologists contains articles on many of the points above.*

## Software and code

Policy information about availability of computer code

| Data collection | Pollen and geochronological data used in these analyses are open access and cited in the manuscript. Data were obtained from the Neotoma Paleoecological Database. All analyses were conducted using the R programming language, version 4.2.2 and all code can be accessed via the Zenodo link. |
|---|---|
| Data analysis | All analyses were conducted using the R programming language, version 4.2.2 and all code can be accessed via the Zenodo link. |

For manuscripts utilizing custom algorithms or software that are central to the research but not yet described in published literature, software must be made available to editors and reviewers. We strongly encourage code deposition in a community repository (e.g. GitHub). See the Nature Portfolio guidelines for submitting code & software for further information.

## Data

Policy information about availability of data

All manuscripts must include a data availability statement. This statement should provide the following information, where applicable:
- Accession codes, unique identifiers, or web links for publicly available datasets
- A description of any restrictions on data availability
- For clinical datasets or third party data, please ensure that the statement adheres to our policy

All pollen and chronological data used in this study are open access and are referenced in the manuscript. Supplementary Table 5 details the pollen sites, and their locations, included in these analyses.

# Research involving human participants, their data, or biological material

Policy information about studies with [human participants or human data](). See also policy information about [sex, gender (identity/presentation), and sexual orientation]() and [race, ethnicity and racism]().

| | |
|---|---|
| Reporting on sex and gender | *Use the terms sex (biological attribute) and gender (shaped by social and cultural circumstances) carefully in order to avoid confusing both terms. Indicate if findings apply to only one sex or gender; describe whether sex and gender were considered in study design; whether sex and/or gender was determined based on self-reporting or assigned and methods used.*<br>*Provide in the source data disaggregated sex and gender data, where this information has been collected, and if consent has been obtained for sharing of individual-level data; provide overall numbers in this Reporting Summary. Please state if this information has not been collected.*<br>*Report sex- and gender-based analyses where performed, justify reasons for lack of sex- and gender-based analysis.* |
| Reporting on race, ethnicity, or other socially relevant groupings | *Please specify the socially constructed or socially relevant categorization variable(s) used in your manuscript and explain why they were used. Please note that such variables should not be used as proxies for other socially constructed/relevant variables (for example, race or ethnicity should not be used as a proxy for socioeconomic status).*<br>*Provide clear definitions of the relevant terms used, how they were provided (by the participants/respondents, the researchers, or third parties), and the method(s) used to classify people into the different categories (e.g. self-report, census or administrative data, social media data, etc.)*<br>*Please provide details about how you controlled for confounding variables in your analyses.* |
| Population characteristics | *Describe the covariate-relevant population characteristics of the human research participants (e.g. age, genotypic information, past and current diagnosis and treatment categories). If you filled out the behavioural & social sciences study design questions and have nothing to add here, write "See above."* |
| Recruitment | *Describe how participants were recruited. Outline any potential self-selection bias or other biases that may be present and how these are likely to impact results.* |
| Ethics oversight | *Identify the organization(s) that approved the study protocol.* |

Note that full information on the approval of the study protocol must also be provided in the manuscript.

# Field-specific reporting

Please select the one below that is the best fit for your research. If you are not sure, read the appropriate sections before making your selection.

☐ Life sciences    ☐ Behavioural & social sciences    ☒ Ecological, evolutionary & environmental sciences

For a reference copy of the document with all sections, see [nature.com/documents/nr-reporting-summary-flat.pdf]()

# Ecological, evolutionary & environmental sciences study design

All studies must disclose on these points even when the disclosure is negative.

| | |
|---|---|
| Study description | An investigation of Holocene (11,700 - 100 calibrated years before present) pollen diversity (richness, evenness, turnover, spatial dissimilarity) trends for all continents except Antarctica, and an examination of how these data relate to a measure of human land use. |
| Research sample | We use pollen and chronological data from data Neotoma and the open-access KK10 Anthropogenic Land Cover Change scenario as a human land use variable. |
| Sampling strategy | We include all pollen records in these analyses if they have at least five samples with 300 pollen grains or more, and three geochronological control points that are sequentially no more than 3000 years apart within the Holocene. |
| Data collection | Pollen data were collected and analysed by individual pollen analysts and archived on the Neotoma Paleoecological Database. |
| Timing and spatial scale | These analyses focus on the Holocene (beginning ~11,700 calibrated years before present) to the present. We truncate our analyses at 100 cal yr BP to prevent surface pollen being included in the analyses. We do not exclude any geographical locations from our analyses, the spatial coverage is determined by the availability of pollen data. |
| Data exclusions | Data inclusion criteria (briefly described above) were pre-established and relate to pollen and geochronological sampling intensity. |
| Reproducibility | All code is available for users to run locally. |
| Randomization | Randomisation is not relevant to this study. |
| Blinding | Blinding is not relevant to our study. |

Did the study involve field work?  ☐ Yes  ☒ No

# Reporting for specific materials, systems and methods

We require information from authors about some types of materials, experimental systems and methods used in many studies. Here, indicate whether each material, system or method listed is relevant to your study. If you are not sure if a list item applies to your research, read the appropriate section before selecting a response.

## Materials & experimental systems

| n/a | Involved in the study |
|-----|-----------------------|
| ☒ ☐ | Antibodies |
| ☒ ☐ | Eukaryotic cell lines |
| ☒ ☐ | Palaeontology and archaeology |
| ☒ ☐ | Animals and other organisms |
| ☒ ☐ | Clinical data |
| ☒ ☐ | Dual use research of concern |
| ☒ ☐ | Plants |

## Methods

| n/a | Involved in the study |
|-----|-----------------------|
| ☒ ☐ | ChIP-seq |
| ☒ ☐ | Flow cytometry |
| ☒ ☐ | MRI-based neuroimaging |

## Plants

Seed stocks
*Report on the source of all seed stocks or other plant material used. If applicable, state the seed stock centre and catalogue number. If plant specimens were collected from the field, describe the collection location, date and sampling procedures.*

Novel plant genotypes
*Describe the methods by which all novel plant genotypes were produced. This includes those generated by transgenic approaches, gene editing, chemical/radiation-based mutagenesis and hybridization. For transgenic lines, describe the transformation method, the number of independent lines analyzed and the generation upon which experiments were performed. For gene-edited lines, describe the editor used, the endogenous sequence targeted for editing, the targeting guide RNA sequence (if applicable) and how the editor was applied.*

Authentication
*Describe any authentication procedures for each seed stock used or novel genotype generated. Describe any experiments used to assess the effect of a mutation and, where applicable, how potential secondary effects (e.g. second site T-DNA insertions, mosiacism, off-target gene editing) were examined.*

