## [Peer Review File · Nature Ecology & Evolution]

Peer Review Information

Journal: Nature Ecology & Evolution

Manuscript Title: Floristic diversity and its relationships with human land use varied regionally during the Holocene

Corresponding author name(s): Jonathan D. Gordon

Editorial Notes:

Reviewer Comments & Decisions:

Decision Letter, initial version:

26th January 2024

Dear Jonny,

[REDACTED] Your manuscript entitled "A spatially extensive multi-metric analysis of Holocene floristic diversity" has now been seen by three reviewers, whose comments are attached. The reviewers have raised a number of concerns which will need to be addressed before we can offer publication in Nature Ecology & Evolution. We will therefore need to see your responses to the criticisms raised and to some editorial concerns, along with a revised manuscript, before we can reach a final decision regarding publication.

In brief, the reviewers recommend adding Neotoma data to ensure better global coverage, implementation of additional metrics, and recommend that you attempt to explicitly examine human impact. We feel editorially that integrating all of these recommendations would help to deliver a substantially stronger paper.

We therefore invite you to revise your manuscript taking into account all reviewer and editor comments. Please highlight all changes in the manuscript text file [OPTIONAL: in Microsoft Word format].

* If you have not done so already please begin to revise your manuscript so that it conforms to our Article format instructions at <http://www.nature.com/natecolevol/info/final-submission>. Refer also to any guidelines provided in this letter.

2* Include a revised version of any required reporting checklist. It will be available to referees (and, potentially, statisticians) to aid in their evaluation if the manuscript goes back for peer review. A revised checklist is essential for re-review of the paper.

[REDACTED]

Nature Ecology & Evolution is committed to improving transparency in authorship. As part of our efforts in this direction, we are now requesting that all authors identified as 'corresponding author' on published papers create and link their Open Researcher and Contributor Identifier (ORCID) with their account on the Manuscript Tracking System (MTS), prior to acceptance. ORCID helps the scientific community achieve unambiguous attribution of all scholarly contributions. You can create and link your ORCID from the home page of the MTS by clicking on 'Modify my Springer Nature account'. For more information please visit www.springernature.com/orcid.

[REDACTED]

Reviewer expertise:

Reviewer #1: palynology, palaeoecology

Reviewer #2: diversity metrics, ecology

Reviewer #3: palynology, palaeoecology

Reviewers' comments:

Reviewer #1 (Remarks to the Author):

2The authors should be congratulated on a generally well-written and thorough article. However, there is a serious lack of transparency when it comes to which sites have been excluded and why. Several key sites from South Africa and Australia seem to have been excluded with no obvious reason why. Maybe a supplementary table can be included listing excluded sites and reason for exclusion. In addition, the data gathering appears to have been carried out some time ago, as more sites are now available from the Australasian region and should be included.

Specific comments:

Line 38: Human-driven declines in what?

Lines 277-282: Very long sentence, consider rephrasing.

Line 616: When was the last update of the Legacy datasets? Neotoma has seen many additions from the Indo-Pacific region over the last 2 years. These would make valuable additions to this work.

Line 617: The Indo-Pacific Pollen Database is also a constituent database in Neotoma and has to be included.

Line 635: How many sites were excluded based on these criteria? There are several key sites missing from Australia (e.g. Lynch's Crater) and South Africa (e.g. Wonderkrater, any Drakensberg sites, etc.).

Line 638: 500 is too high, many researchers choose to "only" count 300 grains. Why has 500 been chosen as the cutoff? Statistical reasons? I appreciate that counts under this amount are still included in separate analyses, but this cutoff is far too high.

Reviewer #2 (Remarks to the Author):

This is an extremely well written, comprehensive study with an important message. It examines long term diversity change through the use of four well known metrics, namely richness, evenness, compositional change and heterogeneity and highlights the historic long term effects of human influence on the environment very nicely. I think this study will have general interest among readers of Nature Ecology and Evolution. However, I do feel that the question and aims could be articulated a little more effectively at an earlier stage in the manuscript and also that the abstract could benefit from the addition of some clarification of question and main message.

The methods (those undertaken prior to the actual diversity measures themselves) are complex and very detailed. As a reader with no background in paleontology I struggled a little to fully understand the steps. The addition of a conceptual or workflow type figure (as an extended figure) would help to guide non experts through the process. I also felt that the study would be more complete were it to include a non abundance based compositional turnover metric such as Jaccard [1]. Use of abundance in compositional diversity measures can affect the results and it would be interesting to see whether

3or not this would hold true here. Either way, the results would be interesting to see as it could either improve the robustness of the Bray Curtis result or suggest that abundance is influential in these analyses.

Minor suggestions

Lines 43-46 suggest the inclusion of Dornelas et al 2019 [2] in support of the balanced gains and losses argument.

Line 186 theres a typo on itra-continental - intra-continental

The addition of a map illustrating the study sites by biome might be nice (as an extended figure)

Figures 2 and 3, Extended Figures 1 and 2

I realise that the vertical grey lines represent the tick marks on the x axis and help to guide the eye to the time periods but maybe they could be a little fainter as I find them a bit distracting from the main result in the plot

1. Jaccard, P. The distribution of the flora in the alpine zone. *New phytologist* 11, 37-50 (1912).

2. Dornelas, M. et al. A balance of winners and losers in the Anthropocene. *Ecology Letters* 22, 847-854, doi:10.1111/ele.13242 (2019).

Reviewer #3 (Remarks to the Author):

The study uses an available global pollen data set (no new data collection, no new data synthesis) and filters for the high quality records and implements state-of-the-art statistical methods to investigate alpha diversity change and rate of change analyses.

The study addresses the question “how the diversity of ecological communities has changed during the deeper history of human development, and whether different metrics of biodiversity change reveal similar or different trends.” Diversity is changing all time, however, the identification of the driver is challenging. As the author specifically targets human impact as a major driver it would have been nice to make use of human indicators contained in pollen records and e.g. related this to diversity changes. Also setting up a testable hypothesis would have been nice. Eg which difference in the diversity changes among the alphasdiversity measure would you apriori expect.

With respect to statistical analyses of diversity and turnover using pollen data the study is generally well implemented. The figures are at a high level. A synthesis figure visualizing the main findings of the paper would have been nice.

4Overall such kind of study is not really new. Several previous studies investigated pollen diversity. For example,

Asia: <https://www.frontiersin.org/articles/10.3389/fevo.2023.1115784/full>

North America: <https://www.pnas.org/doi/10.1073/pnas.2306815120>

Europe: <https://www.nature.com/articles/s41467-019-13233-y>

Global: <https://www.science.org/doi/10.1126/science.abg1685>

line 342: You state that The considerable local variability (Fig. 4) likely reflects diverse human and local and regional climate effects, interacting with local geologies, soils and biotas. This increased turnover can result in increased diversity in some locations but declines in others,....." This interpretation is very unspecific. For example, it is not very clear how you came to the conclusion that the interactions among the different earth system components were causing the difference...why not mainly only one of these factors?

method comment: Pollen records originate from very different archives. It would have been nice to test whether or not this impact diversity trends e.g. I would have expected an impact of source area size ie. some record source the majority of the pollen from few km² while other have a regional to almost subcontinental source.

line 350: Here a direct comparison of the quantitative turnover results would have been nice.

line 379: So what is the new finding from this analysis? The author state: "This highlights the complexity of developing conservation strategies, given that the individual human histories and biodiversity trends of sites, biomes and continents vary, and exclusion of historical human management and other influences may risk reducing rather than increasing local diversity in many instances⁴⁵." While this statement is true, it is not so helpful and it would have been nice to read how this study helped to better understand the complexity.

line 357ff: It would have been nice to bring these information on turnover and alpha diversity change together with real data on land use (either from pollen or other sources).

*****END*****

Author Rebuttal to Initial comments

"Global Holocene floristic diversity changes and human land use", previously "A spatially extensive multi-metric analysis of Holocene floristic diversity"

Nature Ecology and Evolution manuscript number: NATECOLEVOL-23102343

Author responses to referee comments in blue, referee comments in black.

Referees asked for three primary changes:

1) a wider inclusion of pollen records,

2) the addition of an incidence-based beta diversity metric, and

3) an explicit examination of the effects of humans on the observed diversity patterns using either information from pollen indicators or other sources of information on land use change.

We have implemented each of these changes. The changes have facilitated an enriched interpretation particularly through the direct analysis of relationships between the metrics of diversity and those of human land use change. Overall, the manuscript is thereby strengthened, and we greatly appreciate the referees for their suggestions.

Our main changes / additions to the manuscript are as follows:

Principal change 1. Wider inclusion of pollen records. The initial manuscript used pollen data from the LegacyPollen 1.0 data set (which was largely based on records from Neotoma) and associated chronological information from the LegacyAge 1.0 dataset. When they were published (mid-2022), these datasets represented a good spatial and temporal coverage. However, as R1 highlights, many new pollen datasets have since been archived on Neotoma, especially in more data-sparse regions. We have therefore moved away from using the Legacy datasets and now use data downloaded directly from Neotoma. We downloaded all available data from Neotoma and filtered this full dataset using slightly more inclusive chronological control criteria (three chronological control points, rather than five, still with a maximum 3,000 years between successive control points, see revised *Methods*). This has resulted in an additional ~700 pollen records to the global dataset, with the number of pollen records included in Africa increased by ~six times as a result. This change has confirmed and increased the generality of the

overall conclusions, but resulted in some minor differences to the conclusions from the initial submission. We thank the reviewer for the suggestion.

Principal change 2. Addition of incidence-based beta diversity measure. In the initial manuscript, we quantified temporal (“turnover” in the manuscript) and spatial (“heterogeneity” in the manuscript) beta diversity using the abundance-based Bray-Curtis measure, which reflect differences among communities in both the identities of species present and their relative abundances. This is valuable because many community changes represent differences in relative abundances as well as of the identities of different pollen types; and hence we retain the Bray-Curtis analyses in the main text. R2 points out that it would also be interesting to see whether the conclusions change when using an incidence-based beta diversity measure, specifically suggesting Jaccard as a useful additional option. We did not include it previously for reasons of space (and because the results were quite similar), but agree that readers will find the comparison useful. In the revised manuscript, we implement Jaccard for both the through-time and across-space beta diversity measures used in these analyses (shown in new Extended Data Figures, and summarised in the main text). These Jaccard results support the general conclusions of the paper, though there are some minor differences; for example in Europe the Bray-Curtis measure does not pick out an early-Holocene reduction in heterogeneity whereas the Jaccard does. The addition of these metrics (Extended Data Figs. 2-6) provides further detail and allows for a useful comparison, which has strengthened the manuscript.

Principal change 3. Explicit examination of human impact. R3 suggests that this work would contribute more strongly to the literature if it were to include an explicit examination of the effect of human impact/land use on the observed diversity patterns, and we fully agree. As a result, for each continent and biome-within-continent considered, we investigate the relationship between a measure of human land use - the KK10 Anthropogenic Land Cover Change metric (ALCC; the estimated fraction of human-altered land cover per grid cell annually from 8,000 - 100 cal yr BP) - and each of our diversity measures. We include model outputs from these analyses as two new main figures, Fig. 4 and Fig. 5. We find that, across all continents, ALCC is positively associated with compositional turnover, though there is variation in the associations of ALCC with the other metrics (Fig. 4). In Europe, Asia and Australasia, ALCC is positively associated with richness, for example, although we find a range of relationships across the different diversity metrics and continents. These suggested analyses have significantly enriched and improved the manuscript, and we thank the reviewer for this very helpful suggestion. We have inserted an additional paragraph in the results (lines 253 - 298), additional paragraphs in the methods (lines 243 - 251 and lines 960 - 1,014), Figures 4 & 5 (and Extended Data Figs. 6-9), and adjusted the wording in the

more general conclusions to clarify where links between human-associated land cover change changes are strongest (or not so evident in some instances). We have also edited the title of the paper to reflect our consideration of land use change. Further details are provided in our response to R3.

In the following, we address the referees' other comments in turn and explain how we have addressed them.

Referee 1

The authors should be congratulated on a generally well-written and thorough article.

Thanks.

However, there is a serious lack of transparency when it comes to which sites have been excluded and why. Several key sites from South Africa and Australia seem to have been excluded with no obvious reason why. Maybe a supplementary table can be included listing excluded sites and reason for exclusion. In addition, the data gathering appears to have been carried out some time ago, as more sites are now available from the Australasian region and should be included.

We have adjusted our data inclusion criteria in response to this referee's suggestions in *Methods: Data quality inclusion criteria*, described in points 2 & 3 below. In terms of additional data, we initially used the LegacyPollen 1.0 pollen dataset (which was published July 2022), but new pollen datasets have been archived on the Neotoma paleoecological database since then. Therefore, we have made three changes to address R1's concern and ensure greater data inclusiveness:

1. We now use an updated, recent (November 2023) full download of all pollen records from Neotoma (this full download returned 6,097 pollen records).

2. We filter these pollen records using more inclusive criteria relating to the chronological control sampling intensity of a pollen record. We now require three chronological control points (previously five) for a pollen record's inclusion into the dataset, with sequential dates separated by no more than 3,000 years.
3. The main analyses are now based on (repeated) subsamples of 300 grains (rather than 500 previously; see below), enabling more samples and sites to be included.

Together, these changes have resulted in the addition of ~700 pollen records (see Principal change 1, above), particularly strengthening data representation in poorly represented areas such as Africa and the Indo-Pacific. We thank the reviewers for raising this point.

We specify the pollen record selection criteria and database date release version to ensure transparency and will provide the R-code (via a Zenodo link) to enable others to repeat the analyses on these data and on future updates to databases. It is not realistic to provide a supplementary table explaining why >4,000 pollen records did not meet our stringent criteria for data inclusion (insufficient duration, dating, sample sizes to ensure robust conclusions). It is possible to derive such a list, should other researchers be interested in doing so, using our code and comparing the sets of sites included in the analysis with the total Neotoma data set.

Specific comments:

Line 38: Human-driven declines in what?

Changed to “The human-driven erosion of biological diversity and biotic homogenisation...”

Lines 277-282: Very long sentence, consider rephrasing.

Rephrased.

Line 616: When was the last update of the Legacy datasets? Neotoma has seen many additions from the Indo-Pacific region over the last 2 years. These would make valuable additions to this work.

See Principal change 1 and the first response to R1 above.

Line 617: The Indo-Pacific Pollen Database is also a constituent database in Neotoma and has to be included.

Apologies - we have added text to the acknowledgments to address this.

Line 635: How many sites were excluded based on these criteria? There are several key sites missing from Australia (e.g. Lynch's Crater) and South Africa (e.g. Wonderkrater, any Drakensberg sites, etc.).

Following our download of all pollen records on Neotoma (6,097 records), 1,763 pollen records satisfied our inclusion criteria, which overall results in 4,334 pollen records rejected in total. We now include ~700 further pollen records in these analyses than in the initially submitted manuscript that meet our data duration, numbers of samples and chronological control criteria. If specific, named sites meet these criteria, they are included

Line 638: 500 is too high, many researchers choose to "only" count 300 grains. Why has 500 been chosen as the cutoff? Statistical reasons? I appreciate that counts under this amount are still included in separate analyses, but this cutoff is far too high.

Our initial main analysis used a minimum grain count of 150, with a sensitivity of the trends examined using a grain count of 500. In hindsight, 150 grains was too low and 500 grains was too high, so we now

10run the main analyses using a pollen sample filter of 300 grains and we have removed the 500 grain sensitivity analyses. Pollen counts of between 300 and 500 grains have been suggested in the literature (e.g. Keen *et al.*, 2014) for sample inclusion in pollen diversity studies and we therefore feel that this inclusion criterion (300 grains) satisfies the desire to include as many pollen records (sites) as possible, whilst also having sufficient grain counts for meaningful diversity estimates.

Keen, Hayley F., William D. Gosling, Felix Hanke, Charlotte S. Miller, Encarni Montoya, Bryan G. Valencia, and Joseph J. Williams. 'A Statistical Sub-Sampling Tool for Extracting Vegetation Community and Diversity Information from Pollen Assemblage Data'. *Palaeogeography, Palaeoclimatology, Palaeoecology* 408 (2014): 48–59.
<https://doi.org/10.1016/j.palaeo.2014.05.001>.

Referee 2

This is an extremely well written, comprehensive study with an important message. It examines long term diversity change through the use of four well known metrics, namely richness, evenness, compositional change and heterogeneity and highlights the historic long term effects of human influence on the environment very nicely. I think this study will have general interest among readers of Nature Ecology and Evolution.

Many thanks for this positive assessment.

However, I do feel that the question and aims could be articulated a little more effectively at an earlier stage in the manuscript and also that the abstract could benefit from the addition of some clarification of question and main message.

We have now added text in both the introduction and abstract to increase the clarity of the question. In the abstract, we have modified the introductory sentence from:

“Humans have caused growing levels of ecosystem and diversity changes at an increasingly global scale in recent centuries, but longer term changes to diversities are largely unknown.”

to

“Humans have caused growing levels of ecosystem and diversity changes at an increasingly global scale in recent centuries, but longer term diversity trends and how they are affected by human impacts are less well understood.”

In paragraph two of the main text, we also now include the sentence:

“We relate these diversity metrics to estimates of human induced land cover change to evaluate how human land use may have influenced diversity trends over the period of most significant human modification to the Earth’s surface. More specifically, we relate the ‘Krumhardt Kaplan version 2010’^{16,17} (hereafter, ‘KK10’) estimates of anthropogenically induced land cover change (ALCC) to all of our diversity metrics from 8,000 to 100 cal yr BP, the period over which the ALCC estimates and our pollen data overlap.”

We also now add a substantial new section of the abstract that describes the key take-away of the paper. We feel that together, these additional lines clarify the objectives and main message of the paper.

The methods (those undertaken prior to the actual diversity measures themselves) are complex and very detailed. As a reader with no background in paleontology I struggled a little to fully understand the steps. The addition of a conceptual or workflow type figure (as an extended figure) would help to guide non experts through the process.

We appreciate R2's suggestion to include a conceptual workflow figure. We have added this figure as an extended data figure (Extended Data Figure 1), as suggested, and hope that this has made the methods more accessible for non-specialists. Thanks.

I also felt that the study would be more complete were it to include a non abundance based compositional turnover metric such as Jaccard [1]. Use of abundance in compositional diversity measures can affect the results and it would be interesting to see whether or not this would hold true here. Either way, the results would be interesting to see as it could either improve the robustness of the Bray Curtis result or suggest that abundance is influential in these analyses.

Thank you for this suggestion: please see Principal change 2, above. We now include the suggested incidence-based Jaccard as an additional measure to compute temporal compositional turnover and heterogeneity.

Minor suggestions

Lines 43-46 suggest the inclusion of Dornelas et al 2019 [2] in support of the balanced gains and losses argument.

We now include this reference as suggested.

Line 186 theres a typo on itra-continental - intra-continental

Amended.

The addition of a map illustrating the study sites by biome might be nice (as an extended figure)

13The range of helpful analyses by reviewers has led to a substantial increase in the number of main Figures and Extended Data Figures. *Nature Ecology and Evolution* have a limit of ten Extended Data Figures and Tables, all of which we now use. Given this limit, we do not have space for an additional Extended Data Figure showing the pollen records per biome. However, we hope that Fig. 1 (which shows the pollen record locations) plus the new biome maps in Fig. 5 together are sufficient.

Figures 2 and 3, Extended Figures 1 and 2

I realise that the vertical grey lines represent the tick marks on the x axis and help to guide the eye to the time periods but maybe they could be a little fainter as I find them a bit distracting from the main result in the plot

Fair enough - we have removed them and have opted for tick marks instead. Thank you for the suggestion.

Referee 3

Referee 3 suggests that an explicit examination of the effect of changing human impact/land use on the observed diversity patterns would be beneficial. We outline our response to this helpful suggestion in Principal change 3 (above) and below.

The study uses an available global pollen data set (no new data collection, no new data synthesis) and filters for the high quality records and implements state-of-the-art statistical methods to investigate alpha diversity change and rate of change analyses.

The study addresses the question “how the diversity of ecological communities has changed during the deeper history of human development, and whether different metrics of biodiversity change reveal similar or different trends.” Diversity is changing all time, however, the identification of the driver is challenging. As the author specifically targets human impact as a major driver it would have been nice to make use of human indicators contained in pollen records and e.g. related this to diversity changes.

And

line 357ff: It would have been nice to bring these information on turnover and alpha diversity change together with real data on land use (either from pollen or other sources).

We fully agree. In our revised manuscript, we now provide analyses that relate our diversity metrics to regional anthropogenic land cover change reconstructions and added sections to the results, discussion and conclusion, as well as adjusting the title and abstract, that describe these new results.

The diversity metrics are derived from our original analyses (based on the updated and expanded dataset, and adjusted inclusion criteria). We now relate these diversity metrics to anthropogenic land cover change (ALCC) data from the KK10 simulation (Kaplan *et al.*, 2009; 2011) for each continent and biome considered using generalised linear mixed effects models. The KK10 simulation encompasses the time period 8,000 - 100 cal yr BP and provides annual estimates of anthropogenic land cover change as a fraction for each 5 arc-minute (approximately 9km² at the equator) grid cell globally. This encompasses most Holocene land cover change up until 1850 CE. We conducted this analysis at the regional (continental and biome-within-continent) level rather than for each individual site because the ALCC estimates are based on historical population reconstructions and agricultural/pastoral suitability maps for region*time combinations, rather than specific reconstructions for the precise landscape surrounding each site. We present the model fits for each continent in a new Fig. 4 and for individual biomes within Europe and North America in a new Fig. 5.

These figures provide a number of interesting and novel results. They explicitly tie increasing and decreasing diversity in specific places at a global scale to human land use intensity for the first time. Further details are provided in a new section in the *Methods* section, and the results are described in a

15major new section in the main text: *Diversity changes in relation to human land cover changes* (lines 262 - 318).

Kaplan, Jed O., Kristen M. Krumhardt, Erle C. Ellis, William F. Ruddiman, Carsten Lemmen, and Kees Klein Goldewijk. 'Holocene Carbon Emissions as a Result of Anthropogenic Land Cover Change'. *The Holocene* 21, no. 5 (2011): 775–91. <https://doi.org/10.1177/0959683610386983>.

Kaplan, Jed O., Kristen M. Krumhardt, and Niklaus Zimmermann. 'The Prehistoric and Preindustrial Deforestation of Europe'. *Quaternary Science Reviews* 28, no. 27 (2009): 3016–34. <https://doi.org/10.1016/j.quascirev.2009.09.028>.

line 342: You state that The considerable local variability (Fig. 4) likely reflects diverse human and local and regional climate effects, interacting with local geologies, soils and biotas. This increased turnover can result in increased diversity in some locations but declines in others,.....” This interpretation is very unspecific. For example, it is not very clear how you came to the conclusion that the interactions among the different earth system components were causing the difference...why not mainly only one of these factors?

We have reworded our conclusion in this context as a hypothesis. Thousands of ecological studies have stressed the importance of these and additional variables on certain aspects of present-day abundance, species-occurrence and diversity and change, but the data do not exist to partition the relative importance of these on the temporal and spatial scales considered in our paper. Our point was a more general one about heterogeneity (and we gave a few examples to help readers understand what we meant) rather than to apportion relative values to different components of heterogeneity.

line 379: So what is the new finding from this analysis? The author state: “This highlights the complexity of developing conservation strategies, given that the individual human histories and biodiversity trends of sites, biomes and continents vary, and exclusion of historical human management and other influences may risk reducing rather than increasing local diversity in many instances⁴⁵.” While this

16statement is true, it is not so helpful and it would have been nice to read how this study helped to better understand the complexity.

R3's suggestion that we address human impacts allows us to provide increased levels of interpretation. We can now say that the positive association between the turnover of floristic communities and anthropogenic land cover hold across the majority of continents, while increasing pollen richness, evenness and heterogeneity show regionally variable relationships with land use: e.g. positive for Europe, Asia, Australia and previously largely forested biomes in North America, but mainly negative for e.g. Africa, South America, and for North American grasslands and high latitudes. These additional analyses help to uncover some of the complexities of long-term human-flora relationships at a global scale, with a greater focus on the locations for which there is most data (Europe and North America).

See our next response for 'what is new'.

Overall such kind of study is not really new. Several previous studies investigated pollen diversity. For example,

Asia: <https://www.frontiersin.org/articles/10.3389/fevo.2023.1115784/full>

North America: <https://www.pnas.org/doi/10.1073/pnas.2306815120>

Europe: <https://www.nature.com/articles/s41467-019-13233-y>

Global: <https://www.science.org/doi/10.1126/science.abg1685>

These are indeed valuable contributions. However, our manuscript differs from these works in a number of ways.

This is the first paper to simultaneously address turnover (two measures), richness, evenness and spatial beta diversity (two measures) at a global scale, and for European and North American biomes. This is therefore the most spatially extensive study to date, using the most expansive pollen dataset. The only other global-scale Holocene pollen analysis was a rate of change analysis (Mottl *et al.*, 2021), which is related to but not the same as our measures of community turnover, included 1,181 pollen records,

17whereas our revised manuscript includes 1,763 pollen records. Therefore, we provide the first global scale analysis combining 6 metrics of turnover / diversity / heterogeneity, none of which have been reported at this temporal or spatial scale previously.

It also measures turnover and other measures of diversity in the manner adopted by researchers who have been analysing 20th and 21st century biodiversity change. This makes our results for changing patterns and human impacts in the Holocene directly relevant to recent (Anthropocene) recorded biodiversity changes.

With the new valuable analyses suggested by R3, this paper is now also the first to associate pollen diversity changes across all continents with an index of human land use. This work therefore builds on those studies mentioned by R3 to provide an indication of the driver of observed changes, rather than just highlighting patterns.

Mottl, Ondřej, Suzette G. A. Flantua, Kuber P. Bhatta, Vivian A. Felde, Thomas Giesecke, Simon Goring, Eric C. Grimm, et al. 'Global Acceleration in Rates of Vegetation Change over the Past 18,000 Years'. *Science* 372, no. 6544 (2021): 860–64.
<https://doi.org/10.1126/science.abg1685>.

Also setting up a testable hypothesis would have been nice. Eg which difference in the diversity changes among the alphadiversity measure would you apriori expect.

We don't do this explicitly because it would take up a lot of space in the manuscript without really increasing the insights our analyses provided (in this instance, though we thoroughly approve of hypothesis frameworks in other contexts!). We set our paper up more as questions relating to how diversity has changed over longer periods of time and how (in the new analyses) these are related to estimates of human land use change. The null hypothesis is that there is no relationship between time and diversity metrics (Figs. 2, 3) and no relationship between diversity change and land use (Figs. 4, 5);

that continents respond similarly (Figs. 2, 4) and that biomes respond similarly (Figs. 3, 5). This is not particularly interesting to state. In contrast, the alternative hypotheses are too numerous to go through all possible hypotheses that could apply in the context of every combination of diversity metric * continent * biome considered. We are confident, however, that the results we present will generate large numbers of hypotheses that can be tested explicitly by the research community in future.

With respect to statistical analyses of diversity and turnover using pollen data the study is generally well implemented. The figures are at a high level. A synthesis figure visualizing the main findings of the paper would have been nice.

Thank you for these positive words.

In terms of a synthesis figure, we are already maxed-out in terms of the numbers of Figures / Tables included in the paper. Our aspiration in future is to be able to put these results together with other timelines of evidence on diversity change for other taxonomic groups in a graphical manner; but we are not at that point yet.

method comment: Pollen records originate from very different archives. It would have been nice to test whether or not this impact diversity trends e.g. I would have expected an impact of source area size ie. some record source the majority of the pollen from few km² while other have a regional to almost subcontinental source.

We agree that the issues of pollen source area and depositional environment in macro-scale pollen diversity studies are important, though currently there are no studies that directly examine the effects of depositional environment, and associated pollen source areas, on continental and biome-scale pollen diversity patterns (to our knowledge). However, we have addressed R3's concern - and the gap in the literature - by performing a full leave-one-out (~Jackknife) sensitivity analysis on both the temporal diversity patterns and their associations with human land use for all regions (continents and biomes) considered. For each region (continent and biome), we iteratively removed all pollen records from each depositional environment and reran our models on this dataset. There are 82 depositional environments associated with the pollen records contained in our global dataset. We therefore reran our analyses 82

19times and present the results of these leave-one-out analyses in Extended Data Figs 7 & 8 for the continental-scale analyses and Extended Data Figs 7 & 9 for the biome-scale analyses. Across all continents and biomes, temporal diversity trends and their associations with human land use appear robust to depositional environment, and therefore the variety of pollen source areas associated with each.

We add a new section in the methods detailing our leave-one-out sensitivity approach (lines 1,041 - 1,094) and two additional sentences in the *Interpreting pollen diversity patterns* section in the main (lines 320- 333).

line 350: Here a direct comparison of the quantitative turnover results would have been nice.

We reflect on the similarities of some of the Holocene and post-1900 patterns of change in the paper. Researchers will undoubtedly now want to address this issue numerically, and we hope that our work will encourage this. However, the data and analytical challenges involved are such that it is beyond the scope of the current project and manuscript.

Decision Letter, first revision:

24th April 2024

Dear Dr. Gordon,

Thank you for submitting your revised manuscript "Global Holocene floristic diversity changes and human land use." (NATECOLEVOL-23102343A). It has now been seen again by the original reviewers and their comments are below. The reviewers find that the paper has improved in revision, and therefore we'll be happy in principle to publish it in Nature Ecology & Evolution, pending minor revisions to satisfy the reviewers' final requests and to comply with our editorial and formatting guidelines.

20If the current version of your manuscript is in a PDF format, please email us a copy of the file in an editable format (Microsoft Word or LaTeX)-- we can not proceed with PDFs at this stage.

[REDACTED]

Reviewer #1 (Remarks to the Author):

The paper is much improved from the previous version and the authors should be commended on the amount of work they put in to revising the manuscript. All my comments have been addressed admirably and I have no further comments or suggestions.

Reviewer #2 (Remarks to the Author):

I am happy that the authors have addressed all my concerns and feel that the manuscript is now more robust and easier to follow. I feel that the three principal changes have greatly strengthened the manuscript and appreciate the work that will have gone into undertaking these new analyses. The conceptual figure addition in Extended Figure 1 is clear and easy to navigate and this has aided my overall understanding of the processes and analyses undertaken - thank you. In addition, I welcome the inclusion of the R code and addition to the original dataset which I think strengthens findings. I also believe that their findings are enhanced by the inclusion of the Jaccard measure which generally reinforce the Bray Curtis results but which also highlight the greater contribution of relative abundance differences in compositional change in North America (in comparison to Europe). The addition of the section discussing diversity change in relation to human impact/land cover change provides an extra element to the work and again I feel that this enhances the overall strength of the work.

Line 792 suggest succussive to successive

Reviewer #3 (Remarks to the Author):

I agree with the revision and have no further comment. Congratulations for the nice job.

Our ref: NATECOLEVOL-23102343A

15th May 2024

Dear Dr. Gordon,

Thank you for your patience as we've prepared the guidelines for final submission of your Nature Ecology & Evolution manuscript, "Global Holocene floristic diversity changes and human land use." (NATECOLEVOL-23102343A). Please carefully follow the step-by-step instructions provided in the attached file, and add a response in each row of the table to indicate the changes that you have made. Please also check and comment on any additional marked-up edits we have proposed within the text. Ensuring that each point is addressed will help to ensure that your revised manuscript can be swiftly handed over to our production team.

****We would like to start working on your revised paper, with all of the requested files and forms, as soon as possible (preferably within two weeks). Please get in contact with us immediately if you anticipate it taking more than two weeks to submit these revised files.****

In recognition of the time and expertise our reviewers provide to Nature Ecology & Evolution's editorial process, we would like to formally acknowledge their contribution to the external peer review of your manuscript entitled "Global Holocene floristic diversity changes and human land use.". For those reviewers who give their assent, we will be publishing their names alongside the published article.

Nature Ecology & Evolution offers a Transparent Peer Review option for new original research manuscripts submitted after December 1st, 2019. As part of this initiative, we encourage our authors to support increased transparency into the peer review process by agreeing to have the reviewer comments, author rebuttal letters, and editorial decision letters published as a Supplementary item. When you submit your final files please clearly state in your cover letter whether or not you would like to participate in this initiative. Please note that failure to state your preference will result in delays in accepting your manuscript for publication.

Cover suggestions

We welcome submissions of artwork for consideration for our cover. For more information, please see

22our guide for cover artwork.

Nature Ecology & Evolution has now transitioned to a unified Rights Collection system which will allow our Author Services team to quickly and easily collect the rights and permissions required to publish your work. Approximately 10 days after your paper is formally accepted, you will receive an email in providing you with a link to complete the grant of rights. If your paper is eligible for Open Access, our Author Services team will also be in touch regarding any additional information that may be required to arrange payment for your article.

Please note that *Nature Ecology & Evolution* is a Transformative Journal (TJ). Authors may publish their research with us through the traditional subscription access route or make their paper immediately open access through payment of an article-processing charge (APC). Authors will not be required to make a final decision about access to their article until it has been accepted. Find out more about Transformative Journals

Authors may need to take specific actions to achieve compliance with funder and institutional open access mandates. If your research is supported by a funder that requires immediate open access (e.g. according to Plan S principles) then you should select the gold OA route, and we will direct you to the compliant route where possible. For authors selecting the subscription publication route, the journal's standard licensing terms will need to be accepted, including <https://www.nature.com/nature-portfolio/editorial-policies/self-archiving-and-license-to-publish>. Those licensing terms will supersede any other terms that the author or any third party may assert apply to any version of the manuscript.

[REDACTED]

[REDACTED]

Reviewer #1:

Remarks to the Author:

The paper is much improved from the previous version and the authors should be commended on the amount of work they put in to revising the manuscript. All my comments have been addressed admirably and I have no further comments or suggestions.

Reviewer #2:

Remarks to the Author:

I am happy that the authors have addressed all my concerns and feel that the manuscript is now more robust and easier to follow. I feel that the three principal changes have greatly strengthened the manuscript and appreciate the work that will have gone into undertaking these new analyses. The conceptual figure addition in Extended Figure 1 is clear and easy to navigate and this has aided my overall understanding of the processes and analyses undertaken - thank you. In addition, I welcome the inclusion of the R code and addition to the original dataset which I think strengthens findings. I also believe that their findings are enhanced by the inclusion of the Jaccard measure which generally reinforces the Bray Curtis results but which also highlights the greater contribution of relative abundance differences in compositional change in North America (in comparison to Europe). The addition of the section discussing diversity change in relation to human impact/land cover change provides an extra element to the work and again I feel that this enhances the overall strength of the work.

Line 792 suggest successive to successive

Reviewer #3:

Remarks to the Author:

I agree with the revision and have no further comment. Congratulations for the nice job.

Final Decision Letter:

6th June 2024

Dear Jonny,

We are pleased to inform you that your Article entitled "Floristic diversity and its relationships with human land use varied regionally during the Holocene", has now been accepted for publication in Nature Ecology & Evolution.

Over the next few weeks, your paper will be copyedited to ensure that it conforms to Nature Ecology and Evolution style. Once your paper is typeset, you will receive an email with a link to choose the appropriate publishing options for your paper and our Author Services team will be in touch regarding any additional information that may be required

24After the grant of rights is completed, you will receive a link to your electronic proof via email with a request to make any corrections within 48 hours. If, when you receive your proof, you cannot meet this deadline, please inform us at rjsproduction@springernature.com immediately.

Due to the importance of these deadlines, we ask you please us know now whether you will be difficult to contact over the next month. If this is the case, we ask you provide us with the contact information (email, phone and fax) of someone who will be able to check the proofs on your behalf, and who will be available to address any last-minute problems. Once your paper has been scheduled for online publication, the Nature press office will be in touch to confirm the details.

Acceptance of your manuscript is conditional on all authors' agreement with our publication policies (see www.nature.com/authors/policies/index.html). In particular your manuscript must not be published elsewhere and there must be no announcement of the work to any media outlet until the publication date (the day on which it is uploaded onto our web site).

Please note that *Nature Ecology & Evolution* is a Transformative Journal (TJ). Authors may publish their research with us through the traditional subscription access route or make their paper immediately open access through payment of an article-processing charge (APC). Authors will not be required to make a final decision about access to their article until it has been accepted. Find out more about Transformative Journals

Authors may need to take specific actions to achieve compliance with funder and institutional open access mandates. If your research is supported by a funder that requires immediate open access (e.g. according to Plan S principles) then you should select the gold OA route, and we will direct you to the compliant route where possible. For authors selecting the subscription publication route, the journal's standard licensing terms will need to be accepted, including <https://www.nature.com/nature-portfolio/editorial-policies/self-archiving-and-license-to-publish>. Those licensing terms will supersede any other terms that the author or any third party may assert apply to any version of the manuscript.

We welcome the submission of potential cover material (including a short caption of around 40 words) related to your manuscript; suggestions should be sent to Nature Ecology & Evolution as electronic files (the image should be 300 dpi at 210 x 297 mm in either TIFF or JPEG format). Please note that

25such pictures should be selected more for their aesthetic appeal than for their scientific content, and that colour images work better than black and white or grayscale images. Please do not try to design a cover with the Nature Ecology & Evolution logo etc., and please do not submit composites of images related to your work. I am sure you will understand that we cannot make any promise as to whether any of your suggestions might be selected for the cover of the journal.

You can generate the link yourself when you receive your article DOI by entering it here: <http://authors.springernature.com/share>.

[REDACTED]

P.S. Click on the following link if you would like to recommend Nature Ecology & Evolution to your librarian <http://www.nature.com/subscriptions/recommend.html#forms>

** Visit the Springer Nature Editorial and Publishing website at www.springernature.com/editorial-and-publishing-jobs for more information about our career opportunities. If you have any questions please click here.**